# OGPO: Sample Efficient Full-Finetuning of Generative Control Policies

**Sarvesh Patil** [$ 1] **Mitsuhiko Nakamoto** [$ 2] **Manan Agarwal** [‡ 1] **Shashwat Saxena** [‡ 1] **Jesse Zhang** [‡ 3]
**Giri Anantharaman** [1] **Cleah Winston** [3] **Chaoyi Pan** [1] **Douglas Chen** [1] **Nai-Chieh Huang** [1] **Zeynep Temel** [1]
**Oliver Kroemer** [1] **Sergey Levine** [2] **Abhishek Gupta** [3 4] **Hongkai Dai** [† 4] **Paarth Shah** [† 4] **Max Simchowitz** [† 1]

## Abstract

Generative control policies (GCPs), such as diffusion- and flow-based control policies, have proved effective parameterizations for robot learning. This work introduces Off-policy Generative Policy Optimization (OGPO), a sample-efficient algorithm for finetuning GCPs that maintains off-policy critics to maximize data reuse and propagate policy gradients through the full generative process of the policy via a modified PPO objective, using critics as the terminal reward. OGPO achieves state-of-the-art performance on manipulation tasks spanning multitask settings, high-precision insertion, and dexterous control. To our knowledge, it is also the only method that can *fine-tune poorly-initialized behavior cloning policies to near full task-success with no expert data in the online replay buffer*, and does so with *few task-specific hyperparameter tuning*. Through extensive investigations, we demonstrate that OGPO drastically outperforms alternative methods on policy steering and learning residual corrections, and identify the key mechanisms behind its performance. We further introduce practical stabilization tricks, including success-buffer regularization and two-sided conservative advantages to mitigate critic over-exploitation across state- and pixel-based settings. Beyond proposing OGPO, we conduct a systematic empirical study of GCP finetuning, identifying the stabilizing mechanisms and failure modes that govern successful off-policy full-policy improvement.

---
[$]Project Lead. [‡]Equal Contribution. [†]Equal Advising. [1]Carnegie Mellon University [2]University of California, Berkeley [3]University of Washington [4]Toyota Research Institute. Correspondence to: Sarvesh Patil <sarveshp@andrew.cmu.edu>, Mitsuhiko Nakamoto <nakamoto@eecs.berkeley.edu>.

*Proceedings of the 43rd International Conference on Machine Learning*, Seoul, South Korea. PMLR 306, 2026. Copyright 2026 by the author(s).

## 1. Introduction

Autonomous acquisition of new skills is an important challenge for modern robot manipulation. While imitation learning via behavior cloning (BC) from human demonstration can enable a robot to learn behaviors across several contexts, performance is typically brittle to subtle changes in tasks and environments. These models rarely exhibit high success rates zero-shot in the diversity of settings encountered in deployment. While this fragility can be remedied through additional data collection, a natural question to ask is - can the robustness of pre-trained imitation learning policies be bolstered autonomously without requiring considerably more manual data collection?

To this end, there has been a strong interest in finetuning pre-trained robotic policies via reinforcement learning (RL), to autonomously improve behavior via self-collected experience. Of particular relevance is the problem of finetuning *Generative Control Policies* (GCPs) - the parametrization of control policies by expressive generative models, such as diffusion or flow models (Chi et al., 2023; Black et al., 2024; Pan et al., 2025). These policies have been extremely effective for modern robotic applications (Zhang and Gienger, 2024; Wolf et al., 2025).

Current methodology for GCP finetuning succumbs to tradeoffs between data efficiency and the extent of policy improvement during training. Approaches focused on sample efficiency combine *off-policy* critic learning, enabling strong experience reuse, with either targeted *partial* finetuning of the GCP, such as steering the initial generation noise or learning residual corrections, or instead use behavior cloning to imitate high-return actions. These approaches learn quickly when the base policy has strong coverage of optimal actions, but struggle with exploring new behavior. On the other hand, methods focused on eliciting maximum final task performance (Lei et al., 2025; Ren et al., 2024; McAllister et al., 2025) use *on-policy* policy gradient updates, which drive aggressive policy improvement at the expense of significantly compromised sample efficiency.

In this work, we propose a new algorithm - OGPO for full-finetuning of expressive GCPs, providing both sample-

efficient and expressive policy updates via data-efficient off-policy reinforcement learning. Following (Ren et al., 2024; Black et al., 2023) **OGPO** views GCP optimization as a bi-level MDP, with a nested inner denoising MDP over the action generation steps of a GCP, and an outer environment dynamics MDP over actions actually executed in the environment. Importantly, in real-robotics tasks, collecting trajectories from the environment MDP is expensive, while generating action trajectories through the denoising MDP is purely computational and therefore cheap.

While direct policy optimization in the unrolled bi-level MDP can be very (environment-)sample inefficient (Ren et al., 2024; Zhang et al., 2025), **OGPO** leverages the asymmetry of sample costs to perform decoupled policy optimization. **OGPO** performs sample-efficient off-policy TD-learning to learn a Q function in the environment dynamics MDP over *expensive* environment samples, while using data-inefficient, but stable on-policy RL updates to extract policies from the inner denoising MDP over *cheap* GCP samples. Doing so allows for an off-policy policy optimization algorithm that is data-efficient (due to TD-learning in the environment dynamics MDP), while being expressive (due to on-policy RL finetuning in the denoising MDP)

Through careful empirical study, we show that **OGPO** is able to achieve both stable and expressive updates for fine-tuning GCPs in challenging robotics tasks. Based on empirical analysis of the shortcomings, we further propose **OGPO+**, an empirically optimized variant that incorporates improvements in test-time optimization such as Best-of-N planning via Q-functions and policy distillation from successful trajectories obtained via online RL. These improvements allow **OGPO+** to achieve state-of-the-art performance on a set of contact-rich simulation environments with varying horizons, degrees of freedom, and precision requirements, while requiring minimal hyperparameter tuning. Surprisingly, we show that **OGPO+** is able to fine-tune policies with *zero expert data* in the policy replay buffer. This is a fundamentally new capability that points towards the future possibility of finetuning models with minimal human data collected in a task-specific manner on deployment. We perform a careful set of analyses to understand the impact of the decoupled optimization central to **OGPO**, and the impact of the design decision made in **OGPO+** - showing the efficacy of full-policy finetuning of GCPs under the right design choices.

## 2. Preliminaries

We formulate our algorithm as a *Markov Decision Process* (MDP) $M_{\text{ENV}} := (S, A, P_0, P, R, \gamma)$, with states $s \in S$, actions $a \in A$, initial state distribution $P_0$, transition probabilities $P$, reward $R$, and discount factor $\gamma \in (0, 1)$. At each timestep $t$, the agent (e.g., robot) observes the

state $s_t \in S$, takes an action $a_t \sim \pi(a_t \mid s_t) \in A$, transitions to the next state $s_{t+1}$ according to $s_{t+1} \sim P(s_{t+1} \mid s_t, a_t)$ while receiving a reward $R(s_t, a_t)$.[1] For the MDP $M_{\text{ENV}}$, we let $\mathbb{E}^\pi$ (resp. $\mathbb{P}^\pi$) denote the expectation (resp. probability distribution) over trajectories $(s_0, a_0, \dots, s_T, a_T)$ with length $T+1$, with initial state distribution $s_0 \sim P_0$ and transition operator $P$. We train a policy to optimize the cumulative discounted return $J(\pi_\theta) = \mathbb{E}^{\pi_\theta}\left[\sum_{t \geq 0} \gamma^t R(s_t, a_t)\right]$. We also recall the Q-function $Q^\pi(s, a) := \mathbb{E}^\pi[\sum_{t \geq 0} \gamma^t R(s_t, a_t) \mid (s_t, a_t) = (s, a)]$ and value function $V^\pi(s) := \mathbb{E}_{a \sim \pi(s)}[Q^\pi(s, a)]$. We apply action chunking (Zhao et al., 2023), where sequences of actions $a_{t:t+h-1}$ are predicted and executed in open-loop. For simplicity, we treat each action chunk as a single action in $M_{\text{ENV}}$, thereby preserving to the standard MDP notation. Thus, for the rest of the paper, $a_t$ **refers to an entire action-chunk**, and rewards are adjusted appropriately (see Appendix A.1).

**On-Policy Policy Gradient Methods.** *Policy gradient* (PG) methods (e.g., REINFORCE (Williams, 1992)) improve policy performance by approximating the gradient of this objective w.r.t. the policy parameters: $\nabla_\theta J(\pi_\theta) = \mathbb{E}^{\pi_\theta}\left[\sum_{t \geq 0} \nabla_\theta \log \pi_\theta(a_t \mid s_t) \, r_t(s_t, a_t)\right]$, where $r_t(s_t, a_t) := \sum_{\tau \geq t} \gamma^\tau R(s_\tau, a_\tau)$ is the discounted future return from time $t$, and $\nabla_\theta \log \pi_\theta(a_t \mid s_t)$ denotes the gradient of the logarithm of the *likelihood* of $(a_t \mid s_t)$. Myriad improvements exist to reduce variance of gradient estimation and accelerate training stability; following (Ren et al., 2024; Zhang et al., 2025), we build on the PPO algorithm (Schulman et al., 2017). PG methods are called *on-policy* because they optimize over the *current* policy distribution, limiting data re-use and sample efficiency.

**Off-Policy Reinforcement Learning.** *Off-policy RL methods* maintain a replay buffer $\mathcal{D}_{\text{roll}} = \{(s_t, a_t, s_{t+1}, r_t, d_t)\}$ consisting of past states $s_t$, actions $a_t$, subsequent states $s_{t+1}$ from the environment transitions, the observed rewards $r_t$, and the done signal $d_t$. The buffer is used to train an ensemble of $M$ critic networks $Q_{\phi_i} : S \times A \to \mathbb{R}$, with parameters $\phi_1, \dots, \phi_M$, such that $Q_{\phi_i}(s_t, a_t)$ evaluates the expected cumulative return $Q^{\pi_{\bar\theta}}(s_t, a_t)$ of action $a_t$ at state $s_t$ under a current *target policy* $\pi_{\bar\theta}$. The critic networks are updated in parallel using the temporal difference loss, which enforces the Bellman consistency equation defined by Q-functions:

$$L_Q(\phi) = \mathbb{E}\left[Q_\phi(s_t, a_t) - \left(r_t + \gamma \cdot Q_{\text{targ}}(s_{t+1}, a_{t+1})\right)\right]^2,$$
$$(2.1)$$

---

[1]In practice, algorithms may be given incomplete or redundant state observation (e.g., via pixel measurements), in which case we can replace $s$ with an observation $o$. This may violate the Markovianity condition in the MDP, but still leads to well-posed algorithms.

where above the expectation $\mathbb{E}$ is taken over $(s_t, a_t, r_t, s_{t+1} \sim \mathcal{B})$ sampled from the replay buffer $\mathcal{B}$, and each $a_{t+1}$ is sampled independently from the current target policy $\pi_{\bar{\theta}}(\cdot \mid s_{t+1})$. To avoid overestimation bias, we set $Q_{\text{targ}}(s, a) = \frac{1}{M} \sum_i Q_i$ to be a mean over critic networks, described in the Appendix (Appendix A.1). (Fujimoto et al., 2018; Chen et al., 2021).Importantly, (2.1) enables data collected by policies from previous training epochs, thereby increasing sample efficiency.

**Generative Control Policies.** Current robotic control policies use generative models as parameterizations of control policies. Following (Pan et al., 2025), we call these generative control policies (GCPs). GCPs represent a stochastic policy $\pi_{\theta}(\cdot \mid s)$ as a series of iterative computation steps, defined by a mapping $\bar{\pi}_{\theta} : S \times A \times \mathbb{N}$. Given a state $s_t$, the policy first samples $a_{t,K} \sim \bar{\pi}_{\theta}(\cdot \mid a_{t,k} = \emptyset, k = K, s_t)$ where $k$ is a GCP timestep. Next, we sample $a_{t,k-1} \sim \bar{\pi}_{\theta}(\cdot \mid a_{t,k}, k, s_t)$ which leads to an action $a_{t,0}$. We compactly denote the distribution of this action given the observation as $a_{t,0} \sim \pi_{\theta}(\cdot \mid s_t)$, turning the GCP into a standard policy. Our iteration conventions are *decreasing* in $K$, those in diffusion models. Following the same conventions, we also refer to the index $k$ as the "denoising step."

**Flow-Based GCPs.** We focus on a popular class of GCPs: flow-based control policies (Black et al., 2024). As discussed in Appendix C, our methods and baselines can also be instantiated with Diffusion-based policies (Chi et al., 2023) and other controller parametrizations (Pertsch et al., 2025; Frans et al., 2024; Pan et al., 2025) . Flow policies are pretrained using the flow-matching objective: given training pairs $(s, a)$, we sample noise $z \sim \mathcal{N}(0, \mathbf{I})$. With a continuous noise index $\tau \in [0, 1]$, we define an interpolated action $a_{(\tau)} = \tau a + (1 - \tau)z$, and optimize a velocity field $v_{\theta}(a_{(\tau)}, \tau; s)$ by minimizing $\mathbb{E}_{(s,a,\tau)} \| v_{\theta}(a_{(\tau)}, \tau; s) - (a - z) \|^2$ (Albergo et al., 2023; Lipman et al., 2022) . Inference is performed by discretizing an ordinary differential equation (ODE) which reverses the noising process $a_{t,k-1} := a_{t,k} + \frac{1}{K} v_{\theta}(a_{t,k}, k/K, s)$, with $a_{t,0} \sim \mathcal{N}(0, \mathbf{I})$.

### 2.1. Abridged Related Work

Due to space constraints, we describe key related works here and defer a more comprehensive discussion to Appendix I. Generative control policies (GCPs) are based on either diffusion-based (Ho et al., 2020) or flow-based (Lipman et al., 2022) generative models, and have seen widespread adoption in robotic applications (Black et al., 2024; Chi et al., 2023; Bjorck et al., 2025). Given their success, there has been much progress devoted to fine-tuning these policies. Among the most performant *policy-gradient* methods are **DPPO** (Ren et al., 2024) and its adaptation to flow-based policies, ReinFlow (Zhang et al., 2025), which use a two-level MDP formulation described in Section 3.

Off-policy methods, tailored to high sample-efficiency, include: (i) Q-chunking (**QC**, (Li et al., 2025)), which imitates high return actions using best-of-$N$ inference with $Q$ function as verifier (see (3.4)) (ii) Diffusion Steering RL (**DSRL**, (Wagenmaker et al., 2025)), which steers the initial generation noise $a_{t,K}$ in the GCP, (iii) the **EXPO** algorithm (Dong et al., 2025), which learns a *residual* correction to each $a_{t,0}$, and (iv) Policy Agnostic RL (PA-RL (Mark et al., 2024)), which combines **QC** with a gradient ascent step through the $Q$-function during Best-of-$N$ inference. A full description of each method is given in Appendix F.

## 3. Off-Policy Generative Policy Optimization

We propose **O**ff-Policy **G**enerative **P**olicy **O**ptimization, **OGPO**, an off-policy full-policy finetuning method for generative control policies. We begin by introducing the basic algorithm, and then describe an improved variant, **OGPO+**. We provide summary pseudocode in Algorithm 1, and defer full implementation details to Appendix B.

**Background: Off-Policy Policy Extraction.** Given a replay buffer $\mathcal{B} = \{(s, a, s', r)\}$, traditional off-policy RL methods consist of two steps: (1) fitting Q- functions via a TD-update Eq. (2.1), (2) performing policy extraction by choosing actions that maximize the target Q function $Q_{\text{targ}}$ as a surrogate of future return:

$$\theta \in \arg\max_{\theta} \mathbb{E}_{a \sim \pi_{\theta}(s)}(Q_{\text{targ}}(s, a)). \qquad (3.1)$$

The replay buffer facilitates off-policy data-reuse for training $Q_{\text{targ}}$ (typically via (2.1)), driving sample efficiency, whereas (3.1) can be computed purely computationally. A historically popular approach to optimize this objective for simple policy parametrization, like Gaussian policies, is the so-called *reparameterization trick* (Kingma and Welling, 2013; Figurnov et al., 2018), where a stochastic policy is rendered as $\pi_{\theta}(s; w)$ for a noise $w$ drawn from a fixed (non-learned) distribution. From here, Eq. (3.1) is written as an expectation over $w$, the algorithm directly differentiates $Q_{\text{targ}}(s, \pi_{\theta}(s, w))$ with respect to $\theta$ under samples $w$. In principle, the same can be done for GCPs such as flow-policies, sampling an initial noise $a_{t,K}$ and backpropagating through the inference chain (Figure 2, center). However, as we show experimentally (Appendix H.3), doing so leads to an exploding gradient problem as we differentiate through the multiple flow steps. Moreover, it requires differentiating $\nabla_a Q_{\text{targ}}$, which can be inaccurate in contact-rich tasks (Suh et al., 2022).

**OGPO: On-Policy PPO for Off-Policy Policy Extraction.** **OGPO** is designed for applications, such as robotic manipulation, where environment interactions are more costly than computation, and where action gradients with respect to $Q_{\text{targ}}$ are noisy or inaccurate (Suh et al., 2022). We maintain off-policy critic learning that facilitates data reuse,

and propose a *fully parallelizable zero-order optimizer* that solves Eq. (3.1), avoiding backpropagation through the denoising chain and differentiation with respect to the target network.

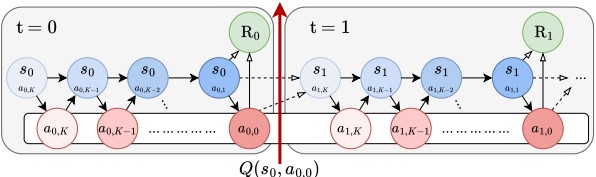

**Figure 1.** We recall the bi-level MDP from (Ren et al., 2024), which embeds action-level trajectories into the environmental dynamics. **OGPO** truncates this MDP at the end of each denoising trajectory, using Q-values as a terminal, action-trajectory-level reward. This enables off-policy policy extraction via on-policy policy optimization.

Our starting point is the bi-level MDP formulation adopted from Ren et al. (2024) (Figure 1). Following (Black et al., 2023), we view sequences $a_{t,K:0} = (a_{t,K}, \dots, a_{t,0})$ as trajectories in an *denoising* MDP, where time is indexed by denoising step $k$, and state and action at step $k$ are $a_{t,k}$ and $a_{t,k-1}$, respectively. Ren et al. (2024) embeds this action-level MDP into the environment-level MDP $M_{\text{ENV}}$, resulting in an *bi-level* MDP where states are $\bar{s}_{t,k} = (s_t, a_{t,k})$, the actions are $a_{t,k-1}$, and the indices $(t, k)$ are lexicographically increasing in $t$ and decreasing in $k$. Figure 1 depicts this bi-level MDP: transitions within each gray block occur within the denoising-level MDP, and between gray blocks are transitions in $M_{\text{ENV}}$; see Appendix D for further details. The **DPPO** algorithm proposed by (Ren et al., 2024) then applies on-policy PPO at the level of this bi-level MDP. Whilst avoiding the aforementioned pathologies associated with backpropagation, this method gives up the sample efficiency afforded by off-policy critic learning.

Our **key insight** is that denoising-trajectories can be generated purely *computationally* from policy inference, as they occur in the "imagination" of the GCP. We can then use critic learning to sever the bi-level MDP just before environment-MDP state transitions (red line, Figure 1), enabling zero-order optimization applied only to the denoising-level MDP. As compared to backpropagation approaches to solving Eq. (3.1), our approach avoids (i) backpropagation through time and (ii) differentiating through the Q-function. Moreover, as compared to pure on-policy zero order optimization through the bi-level MDP (Ren et al., 2024), our zero-order updates are (i) performed purely computationally, in the "imagination" of the denoising process (ii) fully parallelized across large batch sizes (iii) used to optimize a critic network, facilitating full reuse of environment-level trajectories. Moreover, (iv) the problem horizon of the denoising-level MDP scales only with the denoising steps $K$, and not $K \times$ (task horizon).

Concretely, we apply the PPO algorithm (Schulman et al., 2017), a zero-order policy gradient method, to optimize over the denoising MDP. Given state $s_t$, denoising trajectory $a_{t,K:0}$, and baseline value estimate $\hat{V}$, we apply the standard PPO loss *only* to the denoising trajectory $a_{t,K:0}$:

$$\ell_{\text{PPO}}(\theta; s_t, a_{t,K:0}, \hat{\mu}) := \min(\omega_\theta \hat{A}, \text{clip}(\omega_\theta, 1 - \epsilon, 1 + \epsilon)\hat{A})$$

$$\omega_\theta := \prod_{k=1}^{K} \frac{\pi_\theta(a_{t,k-1} \mid s_t, a_{t,k})}{\pi_{\bar{\theta}}(a_{t,k-1} \mid s_t, a_{t,k})}, \ \hat{A} = Q_{\text{targ}}(s_t, a_{t,0}) - \hat{V}. \tag{3.2}$$

Above, $\omega_\theta$ is the likelihood ratio between full denoising trajectories $a_{t,K:0}$ under $\pi_\theta$ and $\pi_{\bar{\theta}}$. The target Q-network $Q_{\text{targ}}$ serves as a terminal reward in the denoising MDP; because no reward is given at intermediate denoising steps, it can equivalently be viewed as a (denoising-level) Monte-Carlo return. The term $\hat{V}$ chosen to approximate $\mathbb{E}_{a_{t,0} \sim \pi_{\bar{\theta}}(s_t)}[Q_{\text{targ}}(s, a_{t,0})]$ in a manner described below. Thus, when $Q_{\text{targ}}$ approximates $Q^{\pi_{\bar{\theta}}}$, $\hat{V}$ approximates the value function $V^{\pi_{\bar{\theta}}}(s_t) = \mathbb{E}_{a_{t,0} \sim \pi_{\bar{\theta}}}[Q^{\pi_{\bar{\theta}}}(s_t, a_{t,0})]$. $\hat{V}$ is called the *baseline term* and renders $\hat{A}$ an approximation of the *advantage function* $A^{\pi_{\bar{\theta}}}(s, a) := Q^{\pi_{\bar{\theta}}}(s, a) - V^{\pi_{\bar{\theta}}}(s, a)$, as desired in policy gradient methods.[2]

**Multiple Denoising-Trajectory Sampling.** Because denoising-trajectories are generated computationally, they can be resampled *fully in parallel* from *any* given state $s_t$ in the replay buffer. Moreover, $Q_{\text{targ}}$ can be evaluated without taking a single transition step in the environment. Taking advantage of this, we evaluate our PPO loss over an average of a batch of parallel-sampled trajectories, purely in the "imagination" of the GCP. By analogy to policy optimization in large language models (LLMs), we can think of a state $s_t$ in the buffer as a "context" and the denoising trajectory $a_{t,K:0}$ as a "response". We draw inspiration from the GRPO algorithm (Shao et al., 2024) in LLM post-training, where multiple responses are sampled in parallel from a given prompt, and gradients are averaged together to reduce gradient variance.[3] In **OGPO**, at each update, we sample $N_{\text{batch}}$ states $(s^{(i)})_{1 \le i \le N_{\text{batch}}}$ from our replay buffer, and sample $N_{\text{group}}$ denoising trajectories $(a_{K:0}^{(i,j)})_{1 \le j \le N_{\text{group}}}$ drawn i.i.d. from $\pi_{\bar{\theta}}(\cdot \mid s^{(i)})$ per state. We then update via the loss

$$\hat{L}_{\text{PPO}}(\theta) = \frac{1}{N_{\text{tot}}} \sum_i \sum_j \ell_{\text{PPO}}(\theta; s^{(i)}, a_{K:0}^{(i,j)}, \hat{V}^{(i)}). \tag{3.3}$$

Eq. (3.3) averages both over the states $s_t^{(i)}$ from the buffer ("prompts"), and denoising-trajectories generated in paral-

---

[2]Subtraction of baselines is particularly essential in the PPO algorithm due to asymmetry induced by the clipping operation.

[3]GRPO includes an additional variance normalization term, which we omit.

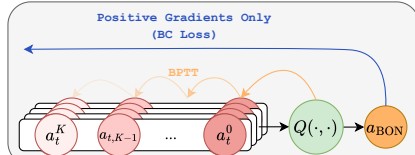
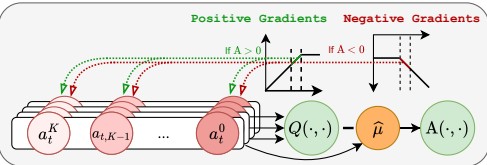

**Figure 2.** Visual depiction of the different off-policy RL algorithms. **(left)** **DSRL** trains an initial noise steering policy, while **EXPO** trains a residual policy to modify the final GCP action. **(center)** **QC** drives policy improvement via supervised finetuning of Best-of-N actions ranked via the critic, while **BPTT** backpropagates the gradients sequentially through the entire GCP. **(right)** **OGPO** uses an ensemble of critics to compute $\hat{A}^{\mathrm{G}}$ (Eq. (3.2)) that update the GCP via Annealed Importance Sampling, thereby directly conditioning the log-likelihoods over the GCP chain.

lel from each given state ("responses"). This yields a normalization factor of $N_{\mathtt{tot}} := N_{\mathtt{batch}} \cdot N_{\mathtt{group}}$. Moreover, parallel sampling facilitates estimating the value baseline via a direct Monte-Carlo approximation $\hat{V}^{(i)} \leftarrow \frac{1}{N_{\mathtt{group}}} \sum_j Q_{\mathtt{targ}}(s^{(i)}, a_0^{(i,j)})$, obviating the need to learn a separate value-prediction network.

**Debiasing Noise Injection for Flow Policies.** We instantiate **OGPO** for flow-based policies. To evaluate the likelihood $\omega_\theta$ in Eq. (3.2), we must ensure the denoising-level action likelihoods $a_{k-1,t} \mid a_{k_t}, s_t$ are non-singular. Rein-Flow (Zhang et al., 2025) modifies the bi-level PPO algorithm of (Ren et al., 2024) for flow-based policies, achieving this by adding additional Gaussian noise to each flow step. For given choice of noise levels $\sigma_k^2$, this yields the following inference procedure:

$$a_{k-1} \sim \bar{\pi}^{\mathrm{FLOW}}(\cdot \mid a_k, k, s) := \mathcal{N}(v_\theta(a_k, \tfrac{k}{K}, s), \sigma_k^2 \mathbf{I})$$

In **OGPO**, we observe that naively adding noise can degrade policy performance by changing the marginal distributions of actions $a_{t,k}$ generated during denoising (Figure 10). We therefore introduce a correction proposed by Albergo et al. (2023) which (in the infinite step limit) ensures the per-denoising-step marginal distributions of noise-augmented actions match those of standard flow sampling; see also (Liu et al., 2025). See Appendix E.2 for details.

### 3.1. **OGPO+**: Best-of-N Inference + Behavior Cloning with a Success Buffer

The core motivation of **OGPO** is to reduce costly interaction time with the environment as much as possible. To this end, we introduce an improved variant, **OGPO+**, which extracts *maximal sample efficiency* through two key modifications.

**1. Best-of-N Inference.** In many domains, such as language modeling, evaluating the quality of an action, or "verification" is learned more quickly and accurately than "generation" of good actions. This verification-generation gap (Setlur et al., 2025) motivates the popular practice of Best-of-$N$ sampling (Brown et al., 2024), where one generates multiple proposal actions, and selects the best using a learned verifier. Best-of-$N$ sampling has seen widespread

---

**Algorithm 1 OGPO** (Abbreviated)

1: **for** each environment step until `done` **do**
2:    Execute $a_t \sim \pi_{\bar{\theta}}(\cdot \mid s_t)$, and update buffer $\mathcal{B} \leftarrow (s_t, a_t, r_t, s_{t+1}, \mathtt{done})$.
      `% Standard Critic Update`
3:    Update critic networks $\phi_1, \dots, \phi_M$ using empirical TD Error (2.1) over $\mathcal{B} \sim \mathcal{D}_{\mathrm{roll}}$.
      `% Actor Update via Multiple Denoising Trajectories`
4:    **for** $i = 1, \dots, N_{\mathtt{batch}}$ **do**
5:       Sample state $s^{(i)}$ from $\mathcal{B}$, and action trajectories $a_{K:0}^{(i,j)} \sim \pi_{\bar{\theta}}(\cdot \mid s^{(i)})$ for $1 \leq j \leq N_{\mathtt{group}}$.
6:       Estimate value baselines via $\hat{V}^{(i)} \leftarrow \frac{1}{N_{\mathtt{group}}} \sum_j Q_{\mathtt{targ}}(s^{(i)}, a_0^{(i,j)})$
7:    **end for**
8:    Update actor using aggregated PPO loss (3.3)
      `% EMA parameters`
9:    Update target parameters $\bar{\theta} \leftarrow (1 - \tau)\bar{\theta} + \tau\theta$, $\bar{\phi}_i \leftarrow (1 - \tau)\bar{\phi}_i + \tau\phi_i$. Set $Q_{\mathtt{targ}} = \frac{1}{N} \sum_i Q_{\bar{\phi}_i}$.
10: **end for**

---

adoption in RL training of robotics policies (Mark et al., 2024; Dong et al., 2025; Li et al., 2025), using the target critic as verifier. **OGPO+** does the same with a slightly modified critic $Q_{\mathrm{BoN}}$ described in Appendix A.1:

$$a_{\mathrm{BoN},t} \leftarrow a_{\mathrm{BoN},t} := \arg\max\{Q_{\mathrm{BoN}}(s_t, a_{t,0}^{(i)}) \mid a_{t,0}^{(1)}, \dots, a_{t,0}^{(N)} \overset{\mathrm{i.i.d}}{\sim} \pi_{\bar{\theta}}(\cdot \mid s_t)\}. \tag{3.4}$$

**2. Behavior cloning regularization from *successful* trajectories.** **OGPO** aggressively optimizes $Q_{\mathtt{targ}}$, which may be an imperfect proxy of task success. The typical "sparse-reward" manipulation setting assigns reward of $-1$ each time step a task remains uncompleted. Thus, minimizing cumulative reward introduces a tension between completion *rate* and completion *speed*. As a result, **OGPO** may attempt to finish tasks too quickly, causing success rates to drop, harming future exploration training stability. We find this occurs with Best-of-$N$ sampling, which further encourages exploitation of $Q_{\mathtt{targ}}$ (see Appendix H.4).

To remedy these challenges, **OGPO+** incorporates a regularization term applied only to actions from successful trajectories. This biases policy improvement toward replicating only the actions that led to success (Oh et al., 2018). Specifically, we maintain a *success buffer* $\mathcal{D}_{\text{succ}} \subseteq \mathcal{D}_{\text{roll}}$ containing transitions from episodes that achieve task success. During training, we sample mini-batches from $\mathcal{D}_{\text{succ}}$ and compute $L_{\text{BC}}(\theta) = \mathbb{E}_{(s_t^{\text{succ}}, a_{t,0}^{\text{succ}}) \sim \mathcal{D}_{\text{succ}}} \left[ \text{BCLOSS}(\bar{\pi}_\theta(\cdot \mid s_t^{\text{succ}}), a_{t,0}^{\text{succ}}) \right]$ where BCLOSS is the appropriate behavior cloning objective (e.g., denoising score matching for diffusion policies, or flow matching loss for flow policies). Success-imitations grounds the policy toward known good actions, while the PPO objective more aggressively explores improvements. For **OGPO+**, the total policy loss combines both terms: $L_{\text{Total}}(\theta) = L_{\text{PPO}}(\theta) + \lambda_{\text{BC}} L_{\text{BC}}(\theta)$.

### 3.2. OGPO+CA: Mitigating the Offline-to-Online Performance Dip via Conservative Advantages

A second challenge in offline-to-online RL is the pervasive "dip" in performance that arises transitioning from offline pretraining to online RL. Warm-starting methods like (Uchendu et al., 2023; Zhou et al., 2024) propose the use of high update-to-data (UTD) ratios or offline datasets during online RL, and the use of pessimistic critic updates. Anecdotally, we find that neither of these methods suffice. Moreover, from Figure 20a, we see that both over- and underestimation of the $Q$ values are possible, and both outliers potentially destabilize training. Thus, we instead to have the policy extraction step maximize the **conservative advantages** . This is made possible because our zero-order extraction takes advantages directly, and also accounts for the fact that global additive errors in critic values are less salient than incorrect *advantage* estimation.

For a given action $a_i$, we set

$$\hat{A}_j^{\text{cons}} = \begin{cases} \min_m A_{j,m} & \text{if } \min_m A_{j,m} > 0, \\ \max_m A_{j,m} & \text{if } \max_m A_{j,m} < 0, , \\ 0 & \text{otherwise.} \end{cases} \quad (3.5)$$

where we recall $A_{j,m} = Q_{\phi_m}(s^{(i)}, a_0^{(i,j)}) - \frac{1}{N_{\text{group}}} \sum_{j=1}^{N_{\text{group}}} Q_{\phi_m}(s^{(i)}, a_0^{(i,j)})$ is the group-wise advantage using the $m$-th network in the ensemble. Eq. (3.5) provides a non-zero advantage (and thus updates the policy) if and only if *all advantages* have the same sign, thereby robustifying updates to estimation errors in the critic networks. As shown in Appendix Figure 20b, we we see that policy extraction with conservative advantages also improves the calibration of critic estimation, in that critic values in earlier states of training more tightly track those in later stages.

## 4. OGPO Evaluation and Comparisons

In this section, we carefully compare **OGPO** to a number of popular baselines to elucidate the merits and limits of its design philosophy—namely full policy fine-tuning, off-policy critic learning, and PPO policy extraction. Our experiment environments are representative of many common challenges in robot learning (e.g. high precision, long horizon, mixed data quality), and baselines cover competing design philosophies (e.g. steering, residual learning).

**Summary of findings.** We summarize comparisons to other off-policy methods in Table 1. Each method has two columns: left denotes if the method converges with task-optimized hyperparameters, and right denotes fixed hyperparameters across all tasks within the criterion (see Appendix K). The markings are explained in the table caption.

With the exception of dense-reward dexterous tasks in the ADROIT environment, we find that **OGPO is able to learn with mixed/partial data quality and on high-precision/long horizon tasks with minimal hyperparameter tuning**, whereas other methods struggle in one or more of these regimes even with hyperparameter tuning. It also exhibits (often times drastic) gains in sample efficiency compared to these methods, and order-of-magnitude improvements related to the on-policy **DPPO** algorithm. Comparisons are detailed further in Section 4.1. Sample efficiency improvements v.s. **DPPO** are expected (off- vs. on-policy), and we attribute gains against off-policy baselines to exploration behavior and expressive policy updates, studied in Section 4.2. Section 4.3 ablates the merits of zero-order vs. backpropagation through time, the role of *negative-advantage gradients* in encouraging exploration, and the enhancements distinguishing **OGPO** and **OGPO+**.

**Baselines.** We compare against the baselines mentioned in Section 2.1, which are described in detail in Appendix F In short, we consider: (i) **DPPO** (Ren et al., 2024), representative of on-policy learning, (ii) **DSRL** (Wagenmaker et al., 2025), representative of off-policy noise steering (iii)

| Criterion | OGPO | | QC | | DSRL | | EXPO | |
|---|---|---|---|---|---|---|---|---|
| Mixed Data Quality | ✓ | ✓ | ✓ | ✓ | ✓ | ✓ | ✗ | ✗ |
| High Precision Tasks | ✓ | ✓ | ✔ | ✓ | ✗ | ✗ | ✓ | ✓ |
| Partial Demonstrations | ✓ | ✓ | ✓ | ✓ | ✔ | ✓ | ✗ | ✗ |
| Long Horizon | ✓ | ✓ | ✓ | ✗ | ✓ | ✗ | ✗ | ✗ |
| Dense/Dexterous | ✔ | ✓ | ✔ | ✓ | ✓ | ✗ | ✓ | ✓ |
| High Sample Efficiency | ✓ | ✓ | ✔ | ✗ | ✗ | ✗ | ✓ | ✗ |

**Table 1.** Left (resp. right) symbol indicates achieving high success With (resp. Without) task-specific hyperparameter tuning. ✗- fails to converge on all tasks; ✓- converges on some but not all tasks; ✔- converges on all tasks, but below SOTA success/efficiency; ✓- converges on all tasks, competitive with SOTA success/efficiency. We use the optimized variants where possible (e.g. **OGPO+** for **OGPO** and similarly for all the baselines).

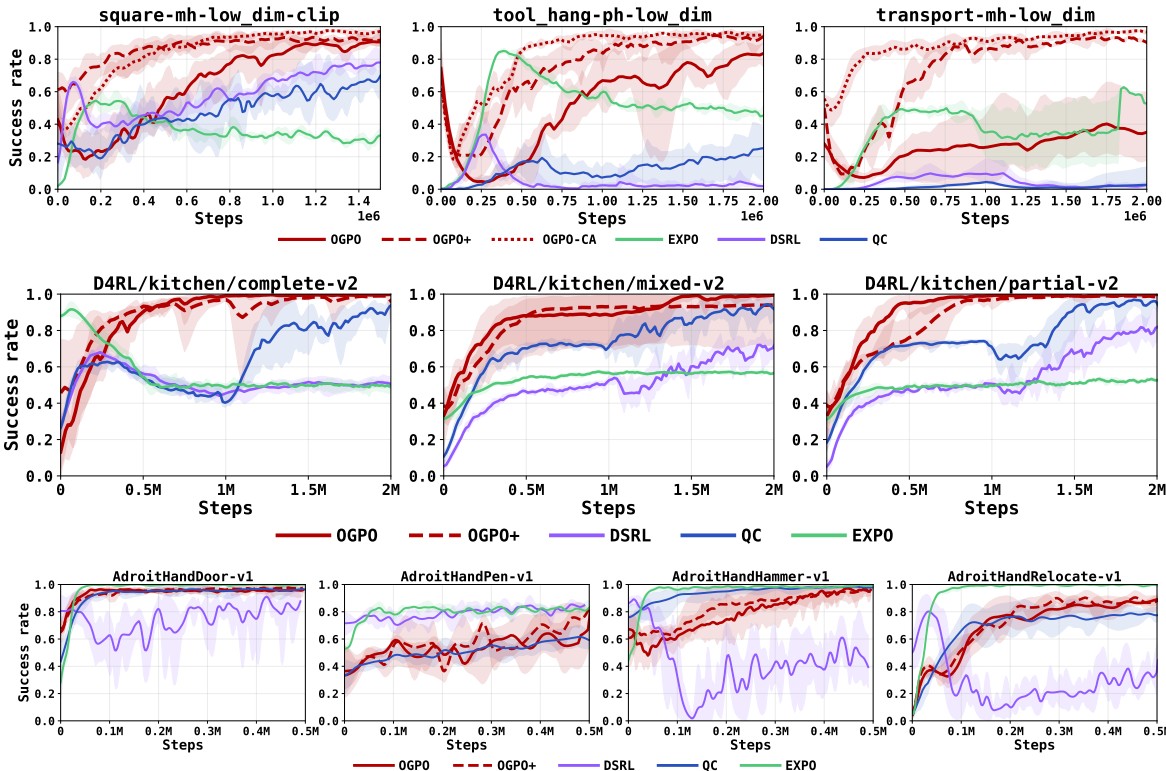

**Figure 3.** Comparison with natural off-policy baselines (**EXPO**, **DSRL**, **QC**) on ROBOMIMIC, FRANKA-KITCHEN, and `AdroitHand`.

**EXPO** (Dong et al., 2025), representative of learning residual corrections to the GCP, and (iv) to a variant of **QC** (Li et al., 2025) representative of behavior cloning policy extraction. We do not compare to ReinFlow (Zhang et al., 2025) due to reported reduced sample efficiency compared to **DPPO**, making the latter a more compelling baseline. We skip comparison to PA-RL (Mark et al., 2024) for reasons described in Appendix F.6. Lastly, we introduce a steering+residual learning baseline, (v) **S/R**, combining **DSRL** and **EXPO** to (hypothetically) yield the benefits of both. Finally, we compare against variants of the above algorithms, **QC+**, **EXPO+**, **DSRL+**, **S/R+**, that incorporate the same optimizations from **OGPO**, when possible. These are also described throughout Appendix F.

**Environments.** Our simulation environments are chosen to elicit key challenges faced in modern robot learning: *Franka Kitchen:* We use the FRANKA-KITCHEN benchmark (Gupta et al., 2019) with a 9-DoF Franka robot manipulating 4 kitchen objects, testing sensitivity to multi-step trajectories with complete demonstrations (KITCHEN-COMPLETE), randomized subtask orders (KITCHEN-MIXED), and partial trajectory data (KITCHEN-PARTIAL). *Robomimic:* To test high-precision robotic control, we use three ROBOMIMIC tasks (Mandlekar et al., 2021): SQUARE (medium-horizon insertion), TOOLHANG (long-horizon multi-step

insertion), TRANSPORT (bi-manual long-horizon transfer). SQUARE and TRANSPORT use Multi-Human (MH) datasets; TOOLHANG uses Proficient-Human (PH) with BC stopped at 50% success. *Adroit:* To test performance in dextrous manipulation tasks with dense-reward, we use the 24-DoF Adroit Hand benchmark: DOOR-V1, HAMMER-V1, PEN-V1, RELOCATE-V1 for door opening, hammering, pen reorientation, and object relocation. Expert datasets from D4RL/Minari. *LIBERO:* Finally, to test pixel-based language-conditioned manipulation, we use the LIBERO benchmark (Liu et al., 2023). Further details are given in Appendix J.

**Experimental Regime: Online RL from a BC Checkpoint.** We emulate real-world robot learning settings where large-scale pretrained policies with varying levels of online success rates are deployed to learn novel tasks without access to the large offline datasets. Thus, we pre-train a flow GCP for all baselines, clip it to at most 50% success rate, and use the same BC checkpoint for all baselines in online RL without additional data. Full details in Appendix K.1.

## 4.1. Comparison to other methods

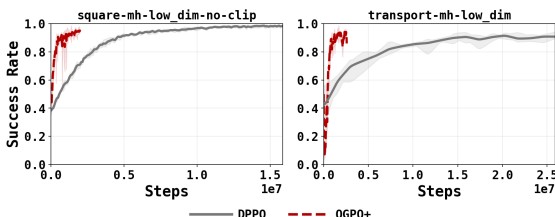

**Figure 4.** **OGPO+** substantially improves sample efficiency compared to the on-policy **DPPO** algorithm .

**Off-Policy OGPO vs. On-Policy DPPO.** We begin by comparing **OGPO+** against **DPPO**, where the major difference between the two is that **OGPO+** truncates the bi-level MDP proposed by **DPPO** at the end of each denoising trajectory with terminal rewards coming from an off-policy Q function, while **DPPO** treats the entire bi-level MDP as a single MDP to train with on-policy RL. On final success rates across two ROBOMIMIC tasks depicted in Figure 4, this off-policy modification results in **DPPO** taking $\sim 7\times$ longer to reach the final success rates achieved by **OGPO+**. Overall, we find that both **OGPO** and **OGPO+** outperform **DPPO**'s paper-reported results in both sample efficiency and final performance across all shared tasks, even with matched network architectures and action chunk lengths.

**Expressivity: Full-Policy Finetuning (OGPO) vs. Steering (DSRL) vs. Residual (EXPO).** Next, we compare **OGPO** to performant off-policy alternatives that do not fine-tune the full GCP across 11 tasks in Figure 3. *Steering* (**DSRL**) can be sample-efficient but relies on sufficient base policy action coverage, leading to suboptimal performance when the base policy's performance is poor, such as in KITCHEN tasks. Further, by not updating later steps of the GCP, steering struggles on high-precision tasks such as the ADROIT task suite. We also empirically found it to be sensitive to hyperparameters; in some tasks, DSRL performance crashes despite heavy tuning. We attribute some of this instability to our use of **DSRL** on a flow-based GCP instead of a diffusion-based GCP; the original paper uses diffusion GCPs for low-data-coverage experiments. However, we are also more sample-efficient than **DSRL**'s paper-reported numbers on shared tasks.

*Residual learning* (**EXPO**) can perform well when the base policy is strong and thus only minor residual corrections are needed (it is highly performant in ADROIT in Figure 3), but, just like steering (**DSRL**), it generally performs poorly or is unstable when the base policy performance starts lower (KITCHEN and most ROBOMIMIC tasks). We note that when given offline data, **EXPO** can perform well (see Figure 19), but our experimental regime is without access to the pre-training data. Our *Steering + Residual Learning* (**S/R**) baseline combines **EXPO** and **DSRL**; we plot sample

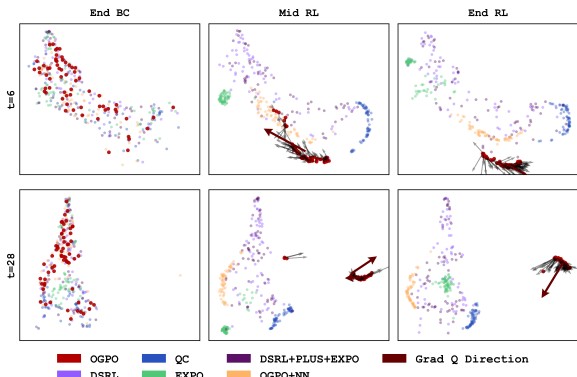

**Figure 5.** We consider 2 critical states (rows) in TOOLHANG and generate a UMAP plot of 64 action samples from **OGPO** and all the baselines combined and plotted separately at the end of BC pretraining, Mid-stage RL checkpoint, and End-stage RL checkpoint for each critical state. We further show $\frac{d\text{UMAP}}{da}\nabla_a Q(s,a)$ vectors at each **OGPO** action to demonstrate the orthogonal conditioning of $\nabla_a Q(s,a)$ to the action spread at critical states.

efficiency curves in ROBOMIMIC tasks in Figure 18, where we see that it is better than **EXPO**/**DSRL** alone in SQUARE, albeit still worse compared to **OGPO**, and demonstrates unstable training in the high precision TOOLHANG task.

**Policy Extraction: PPO (OGPO) vs. Behavior Cloning (BC) with QC.** Finally, we compare *policy extraction* methods. We find the action-chunked **BPTT** variant proposed in the **QC** paper to perform poorly (Fig. 17) on flow policies, and thus use a variant that explores online with Best-of-$N$ action sampling and fine-tunes the BC policy on transitions from the online replay buffer. **QC** plateaus at lower performance for most tasks, requires more task-specific hyperparameter tuning, and has worse sample efficiency. We attribute this to SFT's inability to expand the support of the GCP action distribution, required for sufficient exploration.

### 4.2. Exploration Tendencies of OGPO

As illustrated in Appendix Figure 11, policies trained via Behavior Cloning (BC) exhibit a *mode-seeking* behavior, agnostic of the underlying environment MDP and task rewards. This causes the BC policy to collapse variance rapidly to adhere to narrow "cone" of demonstrated trajectories. In contrast, **OGPO** leverages Q-functions to drive action distribution *manifold expansion* toward high-value regions beyond the support of the offline action distributions.

We ground our understanding of the learning dynamics by visualizing the distributions of actions for the different baselines at certain critical states obtained from TOOLHANG at the end of BC pretraining, and the Initial, Mid, and End stages of RL finetuning. UMAP plots (McInnes et al., 2018) of the combined actions are shown in

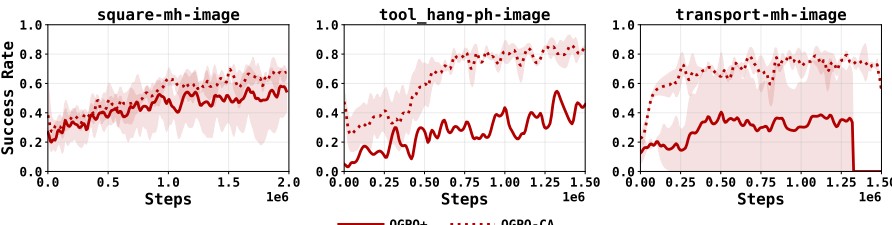

**Figure 6.** **OGPO+** and **OGPO+CA** obviate the need for offline-to-online Q-function RL

**Figure 7.** **OGPO+CA** substantially improves **OGPO**'s training stability in image-based settings with on ROBOMIMIC tasks

Figure 5. We observe a significant overlap of the BC policy actions across all the baselines. The action distances in the UMAP space diverge significantly as training progresses. However, **OGPO** demonstrates action variance reduction in the action space orthogonal to the Q functions as shown by the red arrow(s) in Appendix Figure 13. We empirically show the effects of **OGPO** along with all the baselines on a Push-T task from (Chi et al., 2023) and elucidate a detailed summary of the optimization and exploration behavior of **OGPO** in Appendix G.

### 4.3. Ablations on **OGPO**, **OGPO+**, and **OGPO+CA**

The following ablations are designed to isolate various sub-component decisions within **OGPO** and to explain which design choices align with maximizing sample efficient policy extraction. In the interest of space, full ablation details are given in Appendix H. First, we compare **OGPO**'s zeroth-order policy extraction to Backpropagation Through Time (**BPTT**) that backpropagates first order gradients via Q functions and through the entire GCP denoising chain. As shown in Appendix Figure 17 directly backpropagating through the denoising chain often fails catastrophically, supporting our choice to optimize the GCP via importance sampling rather than through $\nabla_a Q(s,a)$.

Second, using Appendix Figure 15 as reference, Best-of-$N$ inference provides only marginal gains by itself and can increase oscillations when the critic is imperfect. This is consistent with the role of Best-of-$N$ as a verifier of critic learning at inference time, rather as a significant mechanism for policy improvement (Chow et al., 2025; Huang et al., 2025). In contrast, the success buffer used in **OGPO+** consistently improves sample efficiency and asymptotic performance by anchoring

policy improvement to successful behavior. We provide a mathematical basis for the intuition that conditional flow matching (CFM) loss between $\bar{\pi}_\theta$, and the success buffer actions increases the GCP lower-bound on successful modes in Appendix E.3. Moreover, we modify the advantage computation from $\hat{A} = \frac{Q_{\text{targ}}(s_t, a_{t,0}) - \hat{V}}{\hat{\sigma}}$, where $\hat{\sigma}^{(i)} \leftarrow \sqrt{\frac{1}{N_{\text{group}}} \sum_j \left( Q_{\text{targ}}(s^{(i)}, a_0^{(i,j)}) - \hat{V}^{(i)} \right)^2}$ and find that GRPO-style variance normalization hurts performance.

Next, we ablate the offline-to-online Q-learning recipe proposed in Warm Start RL (WSRL, (Zhou et al., 2024)) with and without Calibrated Q-Learning (CalQL, (Nakamoto et al., 2024b)), and compare against OGPO, OGPO+, and OGPO+CA. We find that CalQL+WSRL slightly improves vanilla **OGPO**, but fail to mitigate the policy collapse as prevented by **OGPO+** and **OGPO+CA**. Finally, to validate **OGPO**'s performance on image-based tasks, we ablate **OGPO+** and **OGPO+CA** and further conclude that the stability induced by conservative advantages significantly improves convergence in image-based ROBOMIMIC tasks.

## 5. Conclusion and Limitations

We introduce **OGPO**, an approach that combines the best of on-policy and off-policy methods for fine-tuning generative control policies (GCPs) and enjoys high success rates and sample efficiency across numerous tasks. However, **OGPO** still has drawbacks, the most important being that the parallel denoising rollouts required to estimate Q-values can be prohibitively expensive for large VLA models due to the high inference costs. Future work focusing on Q-function learning fidelity can help ameliorate this limitation by reducing the number of parallel GCP rollouts.

## Impact Statement

This paper presents work whose goal is to advance the field of machine learning. There are many potential societal consequences of our work, none of which we feel must be specifically highlighted here.

## Acknowledgments

MN would like to thank Qiyang Li for helping with the initial implementation, and Zhiyuan Zhou, Seohong Park and Aviral Kumar for their informative discussions. This research used the Savio computational cluster resources provided by the Berkeley Research Computing program at UC Berkeley. MS would like to thank Aviral Kumar and Andrew Wagenmaker for useful discussions. SBP would like to thank Steven Man, Andrea Bajcsy, and Ken Nakamura for their insightful discussions. We acknowledge support from the Toyota Research Institute (TRI) University 2.0 program.

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

# Contents

# A. A Practitioner's Guide to **OGPO**

In this section, we enumerate key design decisions, diagnostic tools, and configurations to serve as a reference for practitioners deploying **OGPO** on new tasks. We defer the pseudocode to Appendix B and the low level hyperparameters to Appendix K

### A.1. Key Design Decisions

While a large set of hyperparameters remain static across all our experiments, some configurations might have a large impact on **OGPO**'s performance on tasks beyond the scope of this paper. We list each item by descending priority level denoted by its high level description followed by the variable name in the official code base.

**0. Action Chunking Conventions and Critic Update**  The main paper denotes each action chunk $a_{t:t+h-1}$ simply as $a_t$ for simplicity. Here we describe how this affects our computation of reward when used to train the resulting Q-function. Let us consider a standard MDP formulation where $s_t$ is the state at current step, and $a_{t:t+h-1}$ denotes the action chunk. We follow the value backup formulation proposed in Q-chunking (Li et al., 2025), where the target uses an $h$-step return over the chunk and bootstraps from the value of the next action chunk at state $s_{t+h}$, with $a_{t+h:t+2h} \sim \pi_\theta(\cdot \mid s_{t+h})$ and $\bar{\theta}$ denoting the parameters of the target network. We use this loss to train the critic for all our off-policy methods, including **OGPO**, **QC**, **DSRL**, and **EXPO**:

$$L_Q(\theta) = \mathbb{E}_{s_t, a_{t:t+h}, s_{t+h} \sim \mathcal{B}} \left[ \left( Q_\theta\left(s_t, a_{t:t+h}\right) - \underbrace{\sum_{t'=1}^{h} \gamma^{t'} r_{t+t'}}_{\text{effective reward}} - \gamma^h Q_{\text{targ}}\left(s_{t+h}, a_{t+h:t+2h}\right) \right)^2 \right]. \tag{A.1}$$

ALGORITHMIC CHOICES

**1. Behavior-cloning regularization from the success buffer (`bc_coeff`).**  The total objective $L_{\text{Total}} = L_{\text{PPO}} + \lambda_{\text{BC}} L_{\text{BC}}$ (Section 3.1) anchors the policy to actions from $\mathcal{D}_{\text{succ}} \subseteq \mathcal{D}_{\text{roll}}$ — the subset of replay-buffer transitions belonging to successful episodes. The regularizer is asymmetric: it raises the likelihood of empirically successful actions but never lowers the likelihood of failed ones, so $L_{\text{BC}}$ contributes a strict lower bound on the modes $L_{\text{PPO}}$ is allowed to abandon. Empirically (Figure 15) this is the single most consequential modification distinguishing **OGPO** from **OGPO+**.In all experiments, we typically select $\lambda = 1.0$.

**2. Conservative advantages (`adv_strategy=conservative`, Eq. (3.5)).**  The conservative advantage $\hat{A}_i^{\text{cons}}$ is non-zero *if and only if all $M$ ensemble members agree on the sign of $A_{j,m}$*, in which case it takes the smallest magnitude consistent with that sign. Two consequences follow: (i) actions on which the ensemble disagrees produce no policy gradient, so the policy is updated only along directions of ensemble consensus; (ii) on directions of consensus, the magnitude is bounded by the most pessimistic Q-function, reducing the impact of outliers in the initial stages of online RL. This significantly mitigates the dip in policy evaluation and yields stable policy extraction.

**3. Critic aggregation for $Q_{\text{targ}}$ and Best-of-$N$(`q_agg`).**  As referenced in Algorithm 5, **OGPO** updates the critic ensemble by minimizing the Temporal Difference (TD) error. To calculate the target values, we employ an ensemble of $M$ target critic networks. The specific method for aggregating these target predictions is determined by the configuration flag `critic_flag`:

$$Q_{\text{targ}}(s', a') = \begin{cases} \min\{Q_{\phi_{i_1}}(s', a'), Q_{\phi_{i_2}}(s', a')\} & \texttt{critic\_flag} = \textbf{subsample} \\ \min_{i \in [M]} Q_{\phi_i}(s', a') & \texttt{critic\_flag} = \textbf{min} \\ \frac{1}{M} \sum_{i=1}^{M} Q_{\phi_i}(s', a') & \texttt{critic\_flag} = \textbf{mean} \end{cases} \tag{A.2}$$

The setting of `critic_flag` is optimized per environment (see 4). The **min** flag uses the minimum all $Q$ networks, which is more aggressively curtails overestimation. The **mean** flag uses the mean, which is less aggressive. Many works have found **subsample** to be a happy medium: we take the minimum of two critic networks whose indices $i_1, i_2$ are

sampled uniformly from the ensemble $\{1, \ldots, M\}$, individually per action. Note that critic training is agnostic to the GCP structure of the policy (Mark et al. (2024)).

Eq. (A.2) aggregates the critic ensemble $\{Q_{\bar{\phi}_m}\}_{m=1}^M$ via $f \in \{\texttt{mean}, \texttt{min}, \texttt{subsample}\}$. Across almost all tasks, we find $\texttt{subsample}$ being the best strategy for $Q_{\text{targ}}$ computation when using synchronous Jax updates, but $\texttt{mean}$ to work best using asynchronous updates. Our experiments are run on using synchronous updates. In both cases, we also find $\texttt{subsample}$ to work optimally for selected the Best-of-$N$ actions Eq. (3.4).

**4. ODE-to-SDE conversion (`error_correct_sde_to_ode`).** In **OGPO**, we add Gaussian noise of standard deviation $\sigma_\tau$ at each flow step to (1) ensure non-singular likelihoods thereby (2) facilitating exploration during online RL. Naively adding isotropic noise to the deterministic update $a_{t,k+1} = a_{t,k} + v_\theta(a_{t,k}, t_k \mid s_t)\Delta t$ causes distribution shift through the denoising chain, so the SDE-inferred policy visits different states than the ODE-inferred policy. Following Albergo et al. (2023), we instead use a marginal path-preserving SDE formulation that adds a score-based drift correction $\frac{\sigma_\tau^2}{2}\nabla \log p_\tau(x_\tau)$. In practice, training a separate score network (as in Liu et al. (2025)) would require modifying the BC pretraining objective, which is prohibitive for pre-trained VLAs. We instead reparameterize the score through the policy and use a tapering noise schedule $\sigma_\tau = \sigma_{\text{init}}\sqrt{1-\tau}$, which avoids the $\tau = 1$ singularity and yields the simple, numerically stable correction term

$$c = \frac{\sigma_{\text{init}}^2 \left( \pi_\theta(x_\tau, \tau)\tau - x_\tau \right)}{2}. \tag{A.3}$$

See Appendix E.2 for the full derivation and Figure 10 for ablations.

**5. Warmup Phase** In the code accompanying **OGPO**, we facilitate an additional *warmup*-phase to pretrain Q-functions. We provide three warmup options:

1. Warm-Start RL (Zhou et al., 2025) with Calibrated Q-Learning (CalQL) (Nakamoto et al., 2024b).

2. Q-function warmup via TD error using $\pi_{\text{BC}}$ rollouts.

3. Q-functions pretrained by regressing MC returns using $\pi_{\text{BC}}$ rollouts.

For the tasks considered in the paper, we generally observe warmup not being critical to policy improvement. The use of Conservative Advantages and SFT via Success Buffer have a much higher impact on **OGPO**'s training stability and sample efficiency.

HYPERPARAMETERS

**1. Group size $N_{\texttt{group}}$ (`grpo_num_samples`).** We rollout $N_{\text{group}}$ trajectories in parallel from a single $s_t$ to compute a mean value estimate for advantage computation in Eq. (3.2). Larger $N_{\text{group}}$ values result in higher exploration and diversity of information points at each update at the cost of compute. We find $N_{\text{group}} = 32$ to be a sweet spot across all our experiments.

**2. PPO clip $\epsilon$ (`clip_epsilon`).** The Annealed Importance Sampling ratio $\omega$ computed in Eq. (3.2) is sensitive to small perturbations in the likelihoods of each denoising step of the GCP being used. For 10-step flow policies, we find a clipping value of $\epsilon = 0.01$ to work best for stable policy extraction. However, practitioners might need to experiment with this ratio depending on their GCP policy parameterization.

**3. Update-to-data ratios** We provide three key update-to-data (UTD) ratios – $\texttt{utd\_warmup}$ (number of critic updates per base policy rollout step), $\texttt{utd\_q}$(number of critic updates per online policy rollout step), and $\texttt{utd\_pi}$(number of actor updates per online policy rollout step). Although a UTD of 1 works across the board, they can be tweaked individually depending on the task setting.

**4. Exponential Moving Average** For all GCP instantiations within **OGPO**, we maintain an Exponential Moving Average (EMA) of the policy weights, denoted as $\theta_{\text{EMA}}$. At every training step, after updating $\theta$, we update $\theta_{\text{EMA}}$ via:

$$\theta_{\text{EMA}} \leftarrow \alpha\theta_{\text{EMA}} + (1-\alpha)\theta, \tag{A.4}$$

where $\alpha$ is a decay rate we typically set $\alpha = 0.995$. For **OGPO**, the EMA serves a dual purpose beyond standard stability. First, it acts as the reference policy $\pi_{\theta_{\text{old}}}$ in the PPO importance sampling ratio (Eq. (3.2)), ensuring that updates are constrained relative to a stable baseline rather than the rapidly changing online policy. Second, for the planning component in **OGPO+**, trajectories for Best-of-N ranking are sampled using $\pi_{\theta_{\text{EMA}}}$ to ensure stability in the candidate actions.

## B. Pseudocode

---

**Algorithm 2 OGPO+**

---

1: $\bar{\pi}_\theta, Q_{\phi_{1\ldots M}}, \mathcal{D}_{\text{roll}} \leftarrow \emptyset, \mathcal{D}_{\text{succ}} \leftarrow \emptyset$
2: $\theta_{\text{targ}} \leftarrow \theta, \phi_{\text{targ}_i} \leftarrow \phi_i \quad \forall i \in \{1, 2, \ldots M\}$
3: **for** iteration $= 1, 2, \ldots$ **do**
4:     Initialize state $s_{t=0} = s_0$ in $M_{\text{ENV}}$
5:     $\mathcal{T}_{\text{ep}} \leftarrow \emptyset$ `Temporary episode buffer`
6:     **while** not `done` **do**
7:         $(s, a, r, s', \texttt{done}) \leftarrow$ TAKE_STEP from the environment
8:         $\mathcal{D}_{\text{roll}} \leftarrow \mathcal{D}_{\text{roll}} \cup \{(s, a, r, s', \texttt{done})\}$
9:         $\mathcal{T}_{\text{ep}} \leftarrow \mathcal{T}_{\text{ep}} \cup \{(s, a, r, s', \texttt{done})\}$
        `% Update critic and policy`
10:         **for** epoch $= 1, 2, \ldots, \texttt{utd}$ **do**
11:           **if** `use_offline` **then**
12:             $\text{B}_{\text{itr}} \sim \{\texttt{r}_{\texttt{offline}}\mathcal{D}_{\text{off}} \cup (1 - \texttt{r}_{\texttt{offline}})\mathcal{D}_{\text{roll}}\}$
13:           **else**
14:             $\text{B}_{\text{itr}} \sim \mathcal{D}_{\text{roll}}$
15:           **end if**
16:           $\text{B}_{\text{succ}} \sim \mathcal{D}_{\text{succ}}$ **if** $\mathcal{D}_{\text{succ}} \neq \emptyset$
17:           UPDATEQ($\text{B}_{\text{itr}}$)
18:           UPDATEGCP($\text{B}_{\text{itr}}, \text{B}_{\text{succ}}$)
          `%Update target networks:`
19:             $\phi_{\text{targ},i} \leftarrow (1 - \tau)\phi_i + \tau\phi_{\text{targ},i} \forall i \in 1, \ldots, M$
20:             $\theta_{\text{targ}} \leftarrow (1 - \tau)\theta + \tau\theta_{\text{targ}}$
21:         **end for**
22:     **end while**
23:     **if** episode successful **then**
24:         $\mathcal{D}_{\text{succ}} \leftarrow \mathcal{D}_{\text{succ}} \cup \mathcal{T}_{\text{ep}}$ $\mathcal{D}_{\texttt{succ}} \subseteq \mathcal{D}_{\texttt{roll}}$
25:     **end if**
26: **end for**
27:
28: **return** converged policy $\pi_\theta$

---

---

**Algorithm 3** Initialization

---

1: **Function** INITIALIZE($\mathcal{D}_{\text{off}}$)
2: {% Policy Initialization}
3: Pre-train GCP $\bar{\pi}_\theta^{\text{BC}}$ on $\mathcal{D}_{\text{off}}$ using BC loss $\mathcal{L}_{\text{BC}}(\theta)$
4: $\bar{\pi}_\theta \leftarrow \bar{\pi}_\theta^{\text{BC}}$
5: {% Critic Initialization}
6: Initialize ensemble of Q functions $Q_{\phi_{1\ldots M}}$
7: **if** `use_calql` **then**
8:     Pre-train $Q_{\phi_{1\ldots M}}$ on $\mathcal{D}_{\text{off}}$ using $\mathcal{L}_{\text{critic}}$ {Optional offline RL}
9: **end if**
10: {% Buffer Initialization}
11: $\mathcal{D}_{\text{roll}} \leftarrow \emptyset$
12: $\mathcal{D}_{\text{succ}} \leftarrow \emptyset$
13: {% Warmup Rollouts}
14: **for** episode = 1, ..., $N_{\text{warmup}}$ **do**
15:     Roll out $\bar{\pi}_\theta^{\text{BC}}$ in $M_{\text{ENV}}$, collect transitions
16:     $\mathcal{D}_{\text{roll}} \leftarrow \mathcal{D}_{\text{roll}} \cup \{(s, a, r, s', \texttt{done})\}_{\text{episode}}$
17:     **if** episode successful **then**
18:         $\mathcal{D}_{\text{succ}} \leftarrow \mathcal{D}_{\text{succ}} \cup \{(s, a, r, s', \texttt{done})\}_{\text{episode}}$
19:     **end if**
20: **end for**
21: **if** `warmup_critic` **then**
22:     **for** step = 1, ..., $N_{\text{critic\_warmup}}$ **do**
23:         $B_{\text{itr}} \sim \mathcal{D}_{\text{roll}}$
24:         UPDATEQ($B_{\text{itr}}$) {Critic-only updates}
25:     **end for**
26: **end if**
27:
28: **return** $\bar{\pi}_\theta, Q_{\phi_{1\ldots M}}, \mathcal{D}_{\text{roll}}, \mathcal{D}_{\text{succ}}$

---

**Algorithm 4** Take A Step In The Environment

---

1: **Function** TAKE\_STEP($s_t$)
2: `done` $\leftarrow$ `False`
3: $a_{t,K} \sim N(0, I)$
4: **for** $k = K, \ldots, 0$ **do**
5:     $a_{t,k-1} \leftarrow \bar{\pi}_{\theta_{\text{targ}}}(a_k, k, s_t)$
6: **end for**
7: $r, s_{t+1} \leftarrow$ Execute $a_{t,0}$ in environment
8: **if** $s_{t+1}$ is terminal **then**
9:     `done` $\leftarrow$ `True`
10: **end if**
11:
12: **return** $(s_t, a_{t,0}, r, s_{t+1}, \texttt{done})$

---

---

**Algorithm 5** Critic Update

---

1: **Function** UPDATEQ($\mathrm{B}_{\text{itr}}$)
2: $(s_t, a_{t,0}, r, s_{t+1}, \texttt{done}) \leftarrow \mathrm{B}_{\text{itr}}$
   With $\theta$ frozen:
3:     $a_{t+1,0} \leftarrow \pi_{\theta_{\text{targ}}}(\cdot \mid s_{t+1})$
4:     $y \leftarrow r + \gamma \cdot \mathbb{I}[\textbf{not } \texttt{done}] \cdot Q_{\text{targ}}(s_{t+1}, a_{t+1,0})$ {Ref. Eq. A.2}
   Update $\phi_{1,\dots,M}$ via gradient descent:
5:     $\nabla_{\phi_i} \frac{1}{|\mathrm{B}_{\text{itr}}|} \sum_{\mathrm{B}_{\text{itr}}} (Q_{\phi_i}(s_t, a_{t,0}) - y)^2$ for $i = 1, \dots, M$

---

---

**Algorithm 6** GCP Update

---

1: **Function** UPDATEGCP($\mathrm{B}_{\text{itr}}$, $\mathrm{B}_{\text{succ}}$)
   On-Policy PPO Update
2:     $s_t \leftarrow \mathrm{B}_{\text{itr}}$
3:     Sample $G$ actions: $\{\bar{\tau}^{(g)}\}_{g=1}^{G} \sim \pi_{\theta_{\text{targ}}}(\cdot \mid s_t)$
4:     $\hat{A}^{\text{G}} = Q_{\text{targ}}(s_t, a_{t,0}^G) - \mu(Q_{\text{targ}}(s_t, a_{t,0}^G))$
5:     $\omega_\theta = \frac{\prod_{k=K}^{0} \bar{\pi}_\theta^{\text{G}}(a_{t,k-1} \mid a_k, k, s_t)}{\prod_{k=K}^{0} \bar{\pi}_{\theta_{\text{targ}}}^{\text{G}}(a_{t,k-1} \mid a_k, k, s_t)}$
6:     $\mathcal{L}_{\text{PPO}}(\theta) = \mathbb{E}_{\bar{\tau} \sim \pi_{\theta_{\text{targ}}}^{\text{G}}} \left[ \min \left( \omega_\theta \cdot \hat{A}^{\text{G}}, \text{clip}(\omega_\theta, 1-\epsilon, 1+\epsilon) \cdot \hat{A}^{\text{G}} \right) \right]$
   BC Update from Success Buffer
7:     $(s_t^{\text{succ}}, a_{t,0}^{\text{succ}}) \leftarrow \mathrm{B}_{\text{succ}}$
8:     $\mathcal{L}_{\text{BC}}(\theta) = \text{BCLoss}(\bar{\pi}_\theta(\cdot \mid s_t^{\text{succ}}), a_{t,0}^{\text{succ}})$ {GCP-specific}
   Combined Update
9:     $\mathcal{L}_{\text{total}}(\theta) = \mathcal{L}_{\text{PPO}}(\theta) + \lambda_{\text{BC}} \mathcal{L}_{\text{BC}}(\theta)$
10: Update $\theta$ via gradient descent on $\mathcal{L}_{\text{total}}(\theta)$

---

## C. Generative Control Policies (GCPs): A Unifying Abstraction

We propose a unifying abstraction for a broad family of popular parameterizations of control policies that we call *Generative Control Policies*, or **GCP** s. GCPs represent a stochastic policy $\pi_\theta(\cdot \mid s)$ as a series of iterative computation steps, defined by a mapping $\bar{\pi}_\theta : S \times A \times \mathbb{N}$. Given a state $s_t$, the policy samples $a_{t,K} \sim \bar{\pi}_\theta(\cdot \mid a_{t,k} = \emptyset, k = K, s_t)$. From then, we sample $a_{t,k-1} \sim \bar{\pi}_\theta(\cdot \mid a_{t,k}, k, s_t)$. The final action proposed is an action $a_{t,0}$. We compactly denote the distribution of this action given the observation as $a_{t,0} \sim \pi_\theta(\cdot \mid s_t)$, turning the GCP into a standard policy. Our iteration conventions are *decreasing* in $K$, following typical convention for diffusion models. We also drop $t$ subscripts when clear from context.

**Examples of GCPs:** In addition to iterative computation, the only other requirement is that the conditional likelihoods, $\log \bar{\pi}_\theta(a_{t,k-1} = a \mid s_t, a_{t,k}, k)$ are efficiently represented. A number of popular parameterizations produce actions iteratively and satisfy this mild requirement:

**Diffusion Policies** (Chi et al., 2023) use Denoising Diffusion Probabilistic Models (DDPMs) Ho et al. (2020). Instantiated as an GCP, these take in pairs $(s, a)$ as training data and iteratively add Gaussian noise to the actions through a forward process $q(a_{k+1} \mid a_k)$ and learn a function $\epsilon_\theta(a_k, k, s)$ predicting the noise added to convert $x_0$ to $x_k$. To produce an action, we sample $a_{t,K} \sim \mathcal{N}(O, \mathbf{I})$, and iteratively generate denoised samples with the following reverse process: $a_{k-1} \sim \bar{\pi}^{\text{DDPM}}(\cdot \mid a_k, k, s) := \mathcal{N}(\mu_k(x_k, \epsilon_\theta(a_k, k, s)), \sigma_k^2 \mathbf{I})$.

**Flow policies** are based on flow matching models. Given training pairs $(s, a)$, we sample noise $z \sim \mathcal{N}(0, \mathbf{I})$, and define the interpolant $a_{(\tau)} := \tau a + (1-\tau)z$ with continuous noise index $\tau \in [0, 1]$. We then learn a velocity field $v_\theta(a_{(\tau)}, \tau, s)$, these predict $\mathbb{E}[a - z \mid s, a_{(\tau)}]$. For $K$ discretization steps, we generate samples by initializing $a_0 \sim \mathcal{N}(0, \mathbf{I})$ and discretizing an ordinary differential equation (ODE) which reverses the noising process $a_{k-1} := a_k + \frac{1}{K} v_\theta(a_k, k/K, s)$ In its stand form, $a_{k-1} \mid a_k, s$ is deterministic. Thus, to convert a flow policy into a proper GCP, for which *likelihoods* are well-defined, we must add additional noise at each step (Reinflow (Zhang et al., 2025)). For a given choice of noise levels $\sigma_k^2$, this induces

the GCP:

$$a_{k-1} \sim \bar{\pi}^{\text{FLOW}}(\cdot \mid a_k, k, s) := \mathcal{N}(v_\theta(a_k, k/K, s), \sigma_k^2 \mathbf{I}) \tag{C.1}$$

While we have presented **OGPO** in the context of *flow-matching* policies, the algorithm is agnostic to the specific generative parameterization of the GCP, and applies directly to diffusion policies as well. Both flow-matching and score-based diffusion policies define an iterative denoising chain $a_t^K \to a_t^{K-1} \to \cdots \to a_t^0$ from a base noise distribution to the action distribution; the only difference is the parameterization of the per-step transition (a learned velocity field $v_\theta$ for flow policies versus a learned score / $\epsilon$-prediction for diffusion). **OGPO**'s key ingredients — per-step likelihood evaluation along the denoising chain (Eq. (3.2)) and the SDE-based exploration noise correction (Appendix E.2) — are derived from generic properties of the underlying SDE and therefore carry over unchanged to a diffusion-policy GCP, provided one substitutes the appropriate noise schedule and score parameterization.

We empirically verify this in Figure 8, where we instantiate **OGPO** on top of a diffusion-policy backbone and observe consistent improvement over BC pretraining, mirroring the trends we report for flow-policy backbones in the main paper. In practice, however, we predominantly default to flow-matching policies for our main experiments: flow policies admit substantially fewer denoising steps at inference time (typically 4–10 versus 50–100 for diffusion) while achieving comparable BC performance, which directly translates into faster environment rollouts and meaningfully reduced wall-clock cost for online RL. We therefore view diffusion-policy **OGPO** as a drop-in alternative whenever the underlying VLA backbone is itself a diffusion model, and flow-policy **OGPO** as the preferred default when inference compute is a bottleneck.

### C.1. OGPO with Diffusion Policies

**OGPO** can, in principle, be combined with any GCPs. Here, as an example, we illustrate its use in diffusion policies. We study this on the SQUARE task, where we pre-train a diffusion policy on the MH dataset and then apply online improvement with **OGPO**. As shown in Figure 8, **OGPO** successfully improves the diffusion policy to achieve mastery.

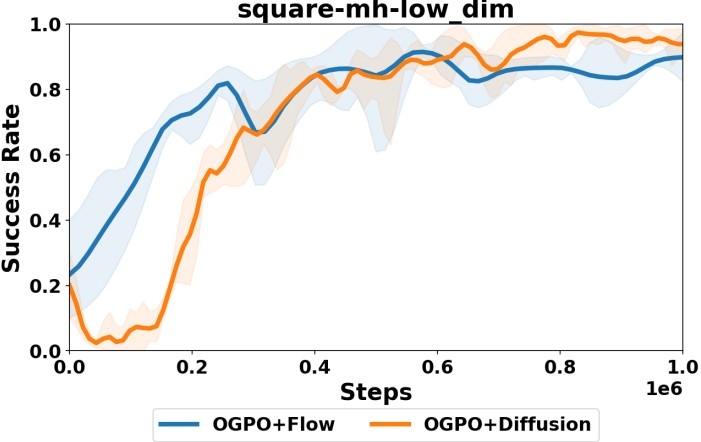

**Figure 8. OGPO with diffusion policies.** **OGPO** can successfully improve both flow policy and diffusion policy.

### C.2. Shortcut Policies

Shortcut policies (Frans et al., 2024) are derived from flow-matching models conditioned on a step-size parameter $d$. The model $\bar{\pi}_\theta(a_t, t, d, o)$ learns to predict the next state of the flow $a_{t+d}$ by taking a shortcut of size $d$. This allows the policy to function as an GCP with a variable number of refinement steps $K$. During pretraining, shortcut models utilize a self-consistency loss that enforces the property that one shortcut step of size $2d$ should be equivalent to two consecutive steps of size $d$:

$$\pi_\theta(a_t, t, 2d, o) \approx \frac{1}{2}\pi_\theta(a_t, t, d, o) + \frac{1}{2}\pi_\theta(a'_{t+d}, t+d, d, o) \tag{C.2}$$

## C.3. Minimal Iterative Policy

Minimal Iterative Policies (MIP) (Pan et al., 2025) represent the simplest GCP instantiation that retains the performance benefits of flow-based policies. The key insight is that the success of generative control policies stems from combining *Stochasticity Injection* during training with *Supervised Iterative Computation*, rather than learning the distributions themselves. MIP uses only $K = 2$ denoising steps, with the first step computing $a_{t,1} \leftarrow \pi_\theta(s_t, a_{t,2} = \bar{0}, t = 0)$, then refining via $a_{t,0} \leftarrow \pi_\theta(s_t, t^\star a_{t,1}, t^\star)$. The core insight being that merely learning the conditional mean is sufficient to match the performance of complex flow-matching policies, provided the refinement steps allow the policy to adhere to the expert action manifold.

Formally, MIP optimizes the following objective during pretraining, where $t^\star = 0.9$ and $z \sim \mathcal{N}(0, I)$ is injected noise:

$$\mathcal{L}_{\text{MIP}}(\theta) = \mathbb{E}\left[\|\pi_\theta(o, I_0 = 0, t = 0) - a\|^2 + \|\pi_\theta(o, I_{t^\star}, t^\star) - a\|^2\right], \tag{C.3}$$

where $I_{t^\star}$ is the interpolant between action $a$ and noise $z$.

## C.4. Tokenized Autoregressive Policies

Tokenized policies, such as those using the FAST tokenizer (Pertsch et al., 2025), represent the action distribution via categorical distributions over a vocabulary of discrete tokens. FAST efficiently handles high-frequency continuous control data by applying a Discrete Cosine Transform (DCT) to action chunks, followed by quantization and Byte-Pair Encoding (BPE).

In this formulation, the GCP is an autoregressive Transformer $\bar{\pi}_\theta(z_k \mid z_{<k}, s_t)$, where $z$ represents the sequence of discrete tokens corresponding to a compressed action chunk. The generative process iteratively samples tokens:

$$z_k \sim \text{Categorical}(\pi_\theta(\cdot \mid z_{<k}, o)) \tag{C.4}$$

Unlike diffusion or flow policies where iteration occurs in continuous action space (refining the values), here iteration occurs in the token sequence space. In particular, this slightly deviates from the GCP formulation described in the main test by requiring conditioning on the whole token sequence $z_{<k}$. However, the light likelihoods in our PPO update in Eq. (3.3) can be easily modified to handle this setting, because $p(z_{1:k}) = \prod_k p(z_k \mid z_{<k})$.

## D. Bi-Level MDP

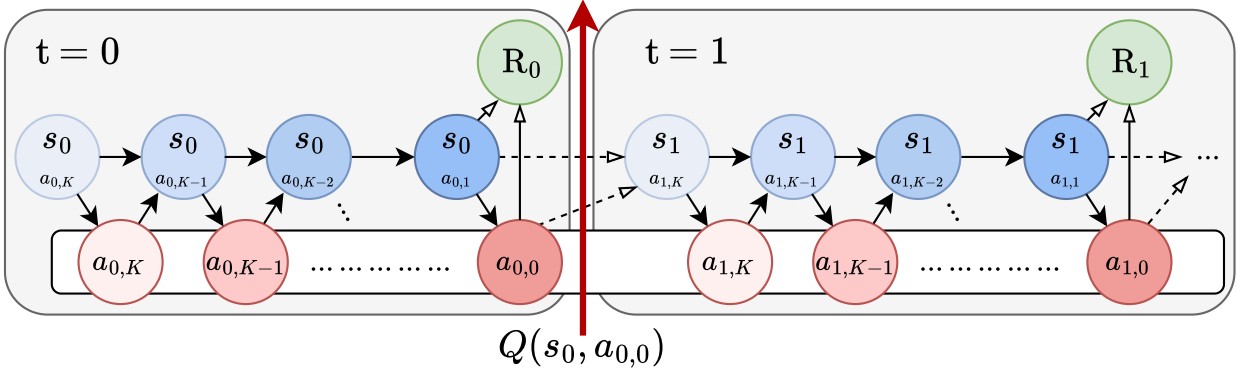

**Figure 9. Bi-level (two-layer) MDP construction.** Each environment step $t$ is expanded into $K$ inner action-generation steps indexed by $k \in \{K - 1, \ldots, 0\}$. The environment transitions and rewards occur only at $k = 0$, while for $k > 0$ the state is unchanged and the inner action variable is updated.

We formulate the bi-level MDP (Figure 9), also called the two-layer MDP in (Ren et al., 2024), by embedding the action-generation dynamics into the environment dynamics. This yields an augmented MDP $M_{\text{BILEVEL}}$ whose trajectory explicitly interleaves environment time with the $K$ action-generation steps.

Recall the environment MDP $M_{\text{ENV}} := (S, A, P_0, P, R, \gamma)$ defined in Section 2. In $M_{\text{BILEVEL}}$, we index time by pairs $(t, k)$, where $t$ denotes the environment step and $k \in \{0, \ldots, K-1\}$ denotes the action-generation step, with $k = 0$ corresponding to executing the final action in the environment. We map $(t, k)$ to a single time index via $\bar{t}(t, k) = tK + (K - k - 1)$, so that the sequence $\bar{t}(t, K-1), \bar{t}(t, K-2), \ldots, \bar{t}(t, 0)$ corresponds to the $K$ generation/execution steps within environment step $t$. The state, action, and reward in $M_{\text{BILEVEL}}$ are defined as

$$\bar{s}_{\bar{t}(t,k)} = (s_t, a_{t,k+1}), \quad \bar{a}_{\bar{t}(t,k)} = a_{t,k}, \quad \bar{R}_{\bar{t}(t,k)}(\bar{s}_{\bar{t}(t,k)}, \bar{a}_{\bar{t}(t,k)}) = \begin{cases} 0, & k > 0, \\ R(s_t, a_{t,0}), & k = 0. \end{cases}$$

Importantly, rewards are emitted only at indices corresponding to executing the environment action, i.e., when $a_{t,0}$ is taken. The initial distribution factorizes as $\bar{P}_0 = P_0 \otimes P_{\text{ACTION},0}$, where $s_0 \sim P_0$ is the initial environment state and $a_{0,K}$ is sampled independently from $P_{\text{ACTION},0}$, the initialization distribution for the action-generation process at $t = 0$.

Finally, the transition kernel is given by

$$\bar{P}(\bar{s}_{\bar{t}+1} \mid \bar{s}_{\bar{t}}, \bar{a}_{\bar{t}}) = \begin{cases} \delta_{(s_t, a_{t,k})}, & \bar{t} = \bar{t}(t, k), \ k > 0 \\ P(\cdot \mid s_t, a_{t,0}) \otimes P_{\text{ACTION},t+1} & \bar{t} = \bar{t}(t, k), \ k = 0 \end{cases},$$

where $P_{\text{ACTION},t}$ (for $t \geq 0$) denotes the initialization distribution for $a_{t,K}$. Intuitively, when $k > 0$, the transition advances the iterative action-generation process by moving from $(s_t, a_{t,k+1})$ to $(s_t, a_{t,k})$ while keeping the environment state fixed; when $k = 0$, it executes $a_{t,0}$ in the environment, samples $s_{t+1} \sim P(\cdot \mid s_t, a_{t,0})$, and re-initializes the next inner process by sampling $a_{t+1,K} \sim P_{\text{ACTION},t+1}$.

# E. Derivations

## E.1. Policy Gradient Loss

An optimal policy parameterized by $\theta$ can be obtained by maximizing an objective function that computes the expected reward over a trajectory $\tau \sim \pi_\theta(\tau)$. Mathematically, $\theta^\star = \arg\max_\theta J(\theta)$, where $J(\theta) = \mathbb{E}_{\tau \sim \pi_\theta(\tau)}[\omega(\tau)]$. Hence, the policy gradient objective is given as:

$$\nabla_\theta J(\theta) = \mathbb{E}_{\tau \sim \pi_\theta(\tau)}[\nabla_\theta \log \pi_\theta(\tau) \omega(\tau)] \tag{E.1}$$

However, there are two main challenges which make the classical PG loss formulation challenging to converge in practice. (1) Policies parameterized as neural networks can only change a little with each gradient step. (2) High variance environments require a very large number of rollouts to obtain $\pi^\star$, which is prohibitively expensive and potentially unsafe to do on real robots. As proposed by (Schulman et al., 2015), high variance can be mitigated by estimating an expectation under a distribution from an older policy $\pi_{\theta_{\text{old}}}$ using importance sampling (IS). This implies use of short horizon replay buffers where actions sampled under $\pi_{\theta_{\text{old}}}$ are reused to compute IS against $\pi_\theta$. This modifies the PG objective as follows:

$$\begin{aligned} \nabla_\theta J(\theta) &= \mathbb{E}_{\tau \sim \pi_\theta(\tau)}\left[\frac{\pi_\theta(\tau)}{\pi_{\theta_{\text{old}}}(\tau)}\nabla_\theta \log \pi_\theta(\tau) \omega(\tau)\right] \\ &= \mathbb{E}_{\tau \sim \pi_\theta(\tau)}\left[\left(\sum_{t=t}^{T}\nabla_\theta \log \pi_\theta(a_t \mid s_t)\right)\left(\prod_{t=1}^{T}\frac{\pi_\theta(a_t \mid s_t)}{\pi_{\text{old}}(a_t \mid s_t)}\right)\left(\sum_{t=t}^{T}r(s_t, a_t)\right)\right] \end{aligned} \tag{E.2}$$

However, the product of importance weights in the trajectory-level estimator leads to vanishing probability products for long horizons $T$. The objective is reformulated using state-action marginals. This separates the expectation over states (dependent on transition dynamics) from the expectation over actions (dependent on the policy):

$$J(\theta) = \sum_{t=1}^{T}\mathbb{E}_{s_t \sim \rho_{\theta_{\text{old}}}(s_t)}\left[\frac{\rho_\theta(s_t)}{\rho_{\theta_{\text{old}}}(s_t)}\mathbb{E}_{a_t \sim \pi_{\theta_{\text{old}}}(\cdot|s_t)}\left[\frac{\pi_\theta(a_t|s_t)}{\pi_{\theta_{\text{old}}}(a_t|s_t)}r(s_t, a_t)\right]\right] \tag{E.3}$$

Calculating the state density ratio $\frac{\rho_\theta(s_t)}{\rho_{\theta_{\text{old}}}(s_t)}$ is difficult as it requires knowledge of the system dynamics. Therefore, TRPO and PPO introduce a simplification by ignoring this term. This results in a biased estimator, but the bias is negligible

provided $\pi_\theta$ remains close to $\pi_{\theta_{\text{old}}}$. The resulting surrogate objective maximizes the probability of actions with high rewards (or advantages) relative to the old policy:

$$J(\theta) \approx \sum_{t=1}^{T} \mathbb{E}_{\substack{s_t \sim \rho_{\theta_{\text{old}}} \\ a_t \sim \pi_{\theta_{\text{old}}}}} \left[ \frac{\pi_\theta(a_t|s_t)}{\pi_{\theta_{\text{old}}}(a_t|s_t)} r(s_t, a_t) \right] \tag{E.4}$$

Classically, algorithms like PPO parameterize the policy $\pi_\theta(a|s)$ as a unimodal Gaussian distribution $\mathcal{N}(\mu_\theta(s), \Sigma)$. This yields a unimodal importance sampling ratio at every timestep $t$, which naturally struggles to model the multimodal action distributions necessary during RL exploration for complex manipulation tasks. Conversely, the total probability $\bar{\pi}_\theta(a_{t,0} \mid s_t)$ in a GCP is the product of the transition probabilities along the generation steps $k$. This likelihood is given as: $\pi_\theta(a_{t,0} \mid s_t) = \prod_{k=1}^{K} \pi_\theta(a_{t,k} \mid s_t)$

Substituting this into the standard PPO objective requires computing the ratio of these products. While trajectory-level importance sampling is unstable for long environment MDP chains (where $T \approx 400 - 1000$), the denoising MDP horizon of the generative process can be sufficiently short (typically $K \leq 10$)

Assuming the current policy $\pi_\theta$ and the reference policy (typically an Exponential Moving Average, $\pi_{\text{EMA}}$) are close, we extend the TRPO formulation to the GCP chains to compute the Annealed Importance Sampling (AIS) ratio:

$$\omega_\theta := \prod_{k=1}^{K} \frac{\pi_\theta(a^{k-1} \mid s, a^k)}{\pi_{\theta_{\text{EMA}}}(a^{k-1} \mid s, a^k)} \tag{E.5}$$

The probability of the final executed action is the joint probability of the entire chain: $\pi_\theta(a_{t,0} \mid s_t) = \prod_{k=K}^{1} \pi(a_{t,k-1} \mid a_{t,k}, s_t)$. We substitute the Monte Carlo return $\omega(\tau)$ with the advantage $\hat{A}$, which yields the final **OGPO**() objective described in Eq. (3.2). When multiplied with the advantage $\hat{A}$, the resulting gradients propagate to every step $k$, updating each in proportion to its contribution to the final action's probability. This end-to-end formulation ensures that generating a high-value action $a_{t,0}$ requires coherent refinement at every step $a_{t,k}$ if the GCP.

### E.2. ODE-to-SDE Exploration Noise Correction

In order to have nondegenerate likelihoods, we ned to convert deterministic flow inference into a stochastic process. Naively, we could add Gaussian noise (as in Zhang et al. (2025)), but the addition of isotropic noise introduces distribution shift between the original action distribution and the noise-augmented action distribution. We note that the same approach is also adopted by (Liu et al., 2025).

Specifically, we follow Albergo et al. (2023), which provides a principled conversion from ODE inference (as in standard flow models) to an SDE). Consider a continuous-time ODE

$$\mathrm{d}X_\tau = v_\theta(X_\tau, \tau)\mathrm{d}\tau, \tag{E.6}$$

where $v_\theta(x_\tau, \tau)$ is the flow velocity field. Next for a time varying diffusion standard deviation $\sigma_\tau$, define an stochastic differential equation (SDE)

$$\mathrm{d}X_\tau^{\text{SDE}} = \underbrace{\left[ v(X_\tau^{\text{SDE}}, \tau) + \frac{\sigma_\tau^2}{2} s(X_\tau^{\text{SDE}}, \tau) \right]}_{v^{\text{SDE}}(X_\tau^{\text{SDE}}, \tau)} \mathrm{d}\tau + \sigma_\tau \, \mathrm{d}W_\tau, \tag{E.7}$$

where $s_\tau(x) = \nabla_x \log \rho_\tau(x)$ is the score function, and where $\rho_\tau$ is the marginal distribution of $X_\tau$. [Albergo et al. (2023)] For every time $\tau$, the marginal distribution of $X_\tau$ and $X_\tau^{\text{SDE}}$ are the same. The key insight is that the correction in the SDE drift $v_\tau^{\text{SDE}} = v_\tau + \epsilon_\tau s_\tau$ directly offsets the effect of the Brownian drift. Furthermore, by Tweedie's formulation, the score function can be computed as

$$s(\tilde{x}_\tau, \tau) = \frac{1}{\sigma}(\mathbb{E}[Z \mid X_\tau + \sigma Z = \tilde{x}_\tau]), \quad Z \sim \mathcal{N}(0, \mathbf{I}) \tag{E.8}$$

In particular, $s_\tau = \frac{1}{\sigma} z_\tau$, where

$$z_\tau \in \arg\min_{z(\cdot)} \mathbb{E} \| z_\tau(X_\tau + \sigma Z) - Z \|^2. \tag{E.9}$$

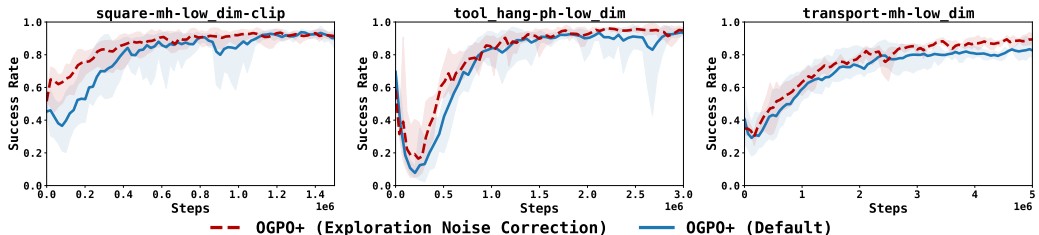

Figure 10. Comparison of ODE-to-SDE correction

**Specialization to OGPO via discretization** Given the SDE with the score correction during online RL:

$$dX_\tau = \left[\bar{\pi}_\theta(x_\tau, \tau) + \underbrace{\frac{\sigma_\tau^2}{2}\nabla \log p_\tau(x_\tau)}_{c}\right] d\tau + \sigma_\tau dW_\tau, \tag{E.10}$$

and noise schedules $\alpha_\tau, \beta_\tau$, the score of the Gaussian probability path $p_\tau(x|z) \sim \mathcal{N}(x_\tau, \alpha_\tau z, \beta_\tau^2 \mathbf{I}_d)$ at timestep $\tau$ is given as

$$\nabla \log p_\tau(x|z) = -\frac{1}{\beta_\tau^2}x + \frac{\alpha_\tau}{\beta_\tau^2}z. \tag{E.11}$$

Reparameterizing the policy wrt the score function gives:

$$\bar{\pi}_\theta(x_\tau, \tau) = \left(\beta_\tau^2 \frac{\dot{\alpha}_\tau}{\alpha_\tau} - \dot{\beta}_\tau \beta_\tau\right)\nabla \log p_\tau(x_\tau) + \frac{\dot{\alpha}_\tau}{\alpha_\tau}x_\tau \tag{E.12}$$

For simplicity, we set $\alpha_\tau = \tau, \beta_\tau = 1 - \tau$. This simplifies Eq. (E.12) to

$$\nabla \log p_\tau(x|z) = \frac{\bar{\pi}_\theta(\mathbf{x}_\tau, \tau)\tau - x}{1 - \tau}. \tag{E.13}$$

Hence, the score correction term $c$ begets

$$c = \frac{\sigma_\tau^2}{2}\nabla \log p_\tau(x|z) \tag{E.14}$$

$$= \frac{\sigma_\tau^2(\bar{\pi}_\theta(\mathbf{x}_\tau, \tau)\tau - x)}{2(1 - \tau)}. \tag{E.15}$$

This reparameterization trick obviates the need for computing score function of the SDE policy, however presents an unstable divide by zero operation at $\tau = 1$, i.e. the last denoising step of the policy in practice. One way to mitigate this is to consider $\alpha_\tau = \tau, \beta_\tau = \sqrt{1 - \tau^2}$ as is done by Liu et al. (2025). However, this requires modification of the BC pretraining objective which is prohibitively expensive for pre-trained VLA models.

Therefore, we instead propose a tapering noise schedule $\sigma_\tau = \sigma_{\text{init}}\sqrt{1 - \tau}$. This results in the score correction term

$$c = \frac{\sigma_{\text{init}}^2(\bar{\pi}_\theta(\mathbf{x}_\tau, \tau)\tau - x)}{2}, \tag{E.16}$$

that prevents numerical instability at the final step of the SDE rollout. We find this tapered noise schedule-based SDE flow policy to be the most stable implementation for **OGPO**. We however note that the runs presented in the paper were generated with a constant noise schedule, but our open sourced codebase provides the most optimal implementation of the SDE-flow policy.

Ablations in Figure 10 reveal the merits of this above sampling correction.

### E.3. BC on $\mathcal{D}_{\text{succ}}$ as an ELBO Barrier in Forward-KL Space

In addition to the policy gradient and pessimism terms described above, **OGPO+** also incorporates a behavior cloning (BC) loss against the success buffer $\mathcal{D}_{\text{succ}}$. We show here that this BC term serves as a tractable lower bound on the forward KL divergence $D_{\text{KL}}(\mathcal{D}_{\text{succ}}\|\pi_\theta)$, thereby aligning $\pi_\theta$ to the modes of successful actions and preventing the policy from dropping their probability mass.

Consider a flow policy $\pi_\theta$ with velocity field $v_\theta(a_\tau, \tau, s)$ trained via the linear interpolant $a_\tau = (1-\tau)\epsilon + \tau a_1$ with target $a_1 - \epsilon$. For any target action distribution $q$, the flow-matching loss admits the bias-variance decomposition

$$\begin{aligned}
\mathcal{L}_{\text{FM}}(\theta; q) &= \mathbb{E}_{\tau,\epsilon,a_1\sim q}\big[\|v_\theta(a_\tau, \tau, s) - (a_1 - \epsilon)\|^2\big] \\
&= \underbrace{\mathbb{E}\big[\|v_\theta - v_q^\star\|^2\big]}_{\theta\text{-optimizable}} + \underbrace{\mathbb{E}\big[\|v_q^\star - (a_1 - \epsilon)\|^2\big]}_{\theta\text{-independent constant } C(q)},
\end{aligned} \tag{E.17}$$

where $v_q^\star(a_\tau, \tau, s) := \mathbb{E}[a_1 - \epsilon \mid a_\tau, \tau, s]$ is the optimal velocity field. By Albergo et al. (2023), integrating $v_q^\star$ via the probability flow ODE in Eq. (E.6) recovers $q$ as the terminal marginal at $\tau = 1$. The first term is therefore a tractable upper bound on the marginal forward KL:

$$\mathcal{L}_{\text{FM}}(\theta; q) - C(q) = D_{\text{KL}}(q \,\|\, \pi_\theta) \geq 0. \tag{E.18}$$

This is an ELBO in the sense that an otherwise-intractable marginal KL — the marginal densities of flow policies have no closed form — is variationally bounded by a tractable squared-error regression loss.

Instantiating this with $q = \mathcal{D}_{\text{succ}}$ recovers the BC loss on the success buffer:

$$\mathcal{L}_{\text{BC}}^{\text{succ}}(\theta) - C(\mathcal{D}_{\text{succ}}) = D_{\text{KL}}(\mathcal{D}_{\text{succ}} \,\|\, \pi_\theta). \tag{E.19}$$

Crucially, the outer expectation is taken under $\mathcal{D}_{\text{succ}}$: every successful action mode is visited at training time. If $\pi_\theta(a_{t,0}^{\text{succ}} \mid s) \to 0$ for some $a_{t,0}^{\text{succ}} \sim \mathcal{D}_{\text{succ}}$, the integrand $\log(\mathcal{D}_{\text{succ}}/\pi_\theta) \to \infty$ and the velocity-MSE penalty pulls $v_\theta$ back toward $v_{\mathcal{D}_{\text{succ}}}^\star$ at that point. This mode-preserving *barrier* property characteristic of forward KL provides regularization via the BC term. Any action mode in $\mathcal{D}_{\text{succ}}$ that $\pi_\theta$ tries to abandon incurs an unbounded penalty. Given the policy gradient conditioning does not strongly pull the GCP distribution against the successful modes, especially in the early training stages, $\pi_\theta$ retains coverage over the full support of successful behaviors throughout online RL.

## F. Baselines

In this section, we describe all baselines we compare to in detail. Throughout, we adopt of the action-chunking conventions of Appendix A.1.

### F.1. Diffusion Policy Policy Optimization (**DPPO**, Ren et al. (2024)

**DPPO** fine-tunes diffusion policies by applying PPO directly to the bi-level MDP introduced in Appendix D. In this construction, each inner denoising step induces an explicit (Gaussian) likelihood, enabling standard policy-gradient updates on the full trajectory in $M_{\text{BILEVEL}}$. **DPPO** then instantiates the PPO clipping objective on $M_{\text{BILEVEL}}$.

Concretely, let $\bar{\pi}_\theta(\bar{a}_{\bar{t}} \mid \bar{s}_{\bar{t}})$ denote the policy on $M_{\text{BILEVEL}}$ (i.e., the diffusion reverse transition at each denoising step). Given trajectories collected from $\bar{\pi}_{\theta_{\text{old}}}$ and advantage estimates $\hat{A}^{\bar{\pi}_{\theta_{\text{old}}}}$, **DPPO** maximizes the PPO clipped surrogate

$$\mathbb{E}_{(s_{\bar{t}}, a_{\bar{t}})\sim\bar{\pi}_{\theta_{\text{old}}}}\left[\min\left(\frac{\bar{\pi}_\theta(a_{\bar{t}} \mid s_{\bar{t}})}{\bar{\pi}_{\theta_{\text{old}}}(a_{\bar{t}} \mid s_{\bar{t}})}\,\hat{A}^{\bar{\pi}_{\theta_{\text{old}}}}(s_{\bar{t}}, a_{\bar{t}}),\, \text{clip}\left(\frac{\bar{\pi}_\theta(a_{\bar{t}} \mid s_{\bar{t}})}{\bar{\pi}_{\theta_{\text{old}}}(a_{\bar{t}} \mid s_{\bar{t}})}, 1-\epsilon, 1+\epsilon\right)\hat{A}^{\bar{\pi}_{\theta_{\text{old}}}}(s_{\bar{t}}, a_{\bar{t}})\right)\right].$$

**DPPO** further uses an advantage estimator tailored to the bi-level structure: since rewards occur only at $\bar{t}(t, 0)$, it computes environment-discounted returns across $t$ and applies an additional denoising discount across $k$ to downweight earlier (noisier) denoising steps.

### F.2. Diffusion Steering Reinforcement Learning (**DSRL**, Wagenmaker et al. (2025)

**DSRL** improves a pretrained diffusion (or flow) policy without updating its weights by learning a policy over the *input noise space* while keeping the denoising dynamics fixed. Whereas a base diffusion policy $\pi_{\text{dp}}$ samples an initial latent $w_t$

from a fixed prior (typically $\mathcal{N}(0, \mathbf{I})$) to maps it to an executed action $a_{t,0}$ via a deterministic denoising chain (e.g., DDIM), **DSRL** instead formulates a *latent-action MDP* in which the fixed prior is replaced by a learnable latent policy $\pi_\psi^{\mathcal{W}}(w_t \mid s_t)$. This policy selects specific noise vectors to steer the frozen denoising process toward actions with higher expected return.

Formally, let $\pi_{\mathrm{dp}}(s_t, w_t)$ denote the action produced by running the (frozen) denoising procedure of $\pi_{\mathrm{dp}}$ initialized at $w_t$, i.e., $a_{t,0} = \pi_{\mathrm{dp}}(s_t, w_t)$. Note that if the denoising sampler is stochastic, interpret $\pi_{\mathrm{dp}}$ as inducing a conditional distribution over $a_{t,0}$ given $(s_t, w_t)$. This induces a latent-action transition kernel

$$P^{\mathcal{W}}(s_{t+1} \mid s_t, w_t) := P\big(s_{t+1} \mid s_t, \pi_{\mathrm{dp}}(s_t, w_t)\big),$$

and **DSRL** optimizes the latent policy by maximizing the discounted return in this latent-action MDP:

$$\max_\psi J(\psi) := \mathbb{E}\left[\sum_{t \geq 0} \gamma^t R\big(s_t, \pi_{\mathrm{dp}}(s_t, w_t)\big)\right], \qquad w_t \sim \pi_\psi^{\mathcal{W}}(\cdot \mid s_t).$$

In practice, $\pi_\psi^{\mathcal{W}}$ is learned with a standard off-policy actor–critic algorithm (e.g., SAC) using transitions $(s_t, w_t, r_t, s_{t+1})$ collected by executing $a_{t,0} = \pi_{\mathrm{dp}}(s_t, w_t)$ in the environment.

**Optimized Variant.** Our **DSRL+** variant applies best-of-N filtering over steering policy actions using the Q-functions and adds a BC-loss using the success buffer to the steering policy on top of the policy graident loss.

### F.3. Expressive Policy Optimization (**EXPO**, Dong et al. (2025)

**EXPO** is designed to stably fine-tune *expressive* pocilices (e.g., diffusion/flow policies) with online RL by avoiding direct value maximization through the expressive policy parameters. Instead, **EXPO** maintains (i) a *base* expressive policy $\pi_{\mathrm{base}}$ trained with a stable imitation (supervised) objective, and (ii) a lightweight Gaussian *edit* policy $\pi_{\mathrm{edit}}$ that performs local action refinement toward higher $Q$-values. At interaction time, **EXPO** constructs an *on-the-fly* (OTF) policy that samples candidate actions from $\pi_{\mathrm{base}}$, refines them with $\pi_{\mathrm{edit}}$, and executes the candidate with the highest critic value; the same OTF selection is also used inside the TD backup.

Given $a \sim \pi_{\mathrm{base}}(\cdot \mid s)$, **EXPO** samples an additive edit $\delta \sim \pi_{\mathrm{edit}}(\cdot \mid s, a)$ and forms the refined action $\tilde{a} = a + \delta$. The OTF policy selects the better of the candidates according to the critic, $a^*(s) \in \arg\max_{a' \in \{a, \tilde{a}\}} Q_\phi(s, a')$. The edit policy is updated to increase the value of refined actions (with entropy regularization).

$$\max_{\pi_{\mathrm{edit}}} \mathbb{E}_{(s,a) \sim \mathcal{D}, \delta \sim \pi_{\mathrm{edit}}} \left[ Q_\phi(s, a + \delta) - \alpha \log \pi_{\mathrm{edit}}(\delta \mid s, a) \right].$$

The critic is trained by TD regression using the same OTF selection for the next-state action: $\min_\phi \mathbb{E}\left[ \big(r + \gamma Q_{\phi'}(s', a^*(s')) - Q_\phi(s, a_t)\big)^2 \right]$. Finally, $\pi_{\mathrm{base}}$ is updated only through imitation-style regression (not direct $Q$-maximization), with value improvement coming from $\pi_{\mathrm{edit}}$ and the OTF selection.

**Improve Variant.** **EXPO+** modifies the behavior cloning term in the standard **EXPO** for the "success buffer" variant described in Section 3.1.

### F.4. Q-Chunking (**QC**, Li et al. (2025))

Recall that, in our notation, we use a single action $a_t$ to decode an entire action-chunk in a the true environment, $a_{t:t+h-1}$. The **QC** algorithm proposes multiple variants. One of which, when specialized to GCPs, would require backpropagation through denoising steps, which we show leads to poor performance in Figure 17. Therefore, we opt for the other variant, which amounts to simply best-of-$N$ inference plus behavior cloning. This variant of **QC** consists of three simple components:

- Learn a critic $Q(s, a)$, following the action-chunking conventions in Appendix A.1. Use this to train the critic via Eq. (2.1).

- Compute the Best-of-$N$ action, by $Q_{\mathrm{targ}}$, as following Eq. (3.4).

- Finally, we use a behavior cloning loss applied to past $(s, a)$ pairs collected by the above planning mechanism,.

**Optimized Variant.** Our **QC+** variant only applies BC loss to successful actions.

### F.4.1. Q-CHUNKING V/S OGPO

Q-Chunking learns Q-functions that evaluate entire action chunks as atomic units, treating $Q(s, a_{1:H})$, where $a_{1:H}$ denotes the full action sequence over a horizon $H$. This formulation is agnostic to how the action chunk is generated—whether via a flow policy, a diffusion model, or direct regression. Policy improvement is guided using the Q-functions to rank a batch of actions and perform supervised fine-tuning (SFT) using BC loss on the Best-of-N actions. In contrast, **OGPO** explicitly leverages the iterative structure of the Generative Control Policy (GCP) by computing annealed importance sampling ratios over the denoising chain Eq. (3.2). Moreover, the advantage computation evaluates the group relative Q values over the entire action chunk and the policy gradient loss propagates through *every* denoising step $k$. This end-to-end formulation ensures that producing a high-value action requires coherent refinement at every GCP step, rather than treating the generation process as a black box.

### F.5. ReinFlow (Zhang et al. (2025), not compared)

The ReinFlow algorithm (Zhang et al., 2025) is nearly identical to **DPPO**, except that it uses a flow policy as a base policy instead of Diffusion. To get non-singular likelihood rations, it augments the flow model with additional noise. However, their reported numbers are less sample efficient than **DPPO** (the flow sampling, however, improves *computational* efficiency), so we only use **DPPO** as a stronger baseline.

### F.6. PA-RL (Mark et al. (2024), not compared)

The PA-RL (Mark et al., 2024) algorithm is similar to QC, but includes an additional gradient ascent step $a' \leftarrow \nabla_a Q(s, a)$ to further improve actions. These gradient computations present a significant computational overhead, and perform best on TPU hardware. We found this method infeasible to run given our compute budget. Furthermore, given the instability of Q-gradients in non-smooth tasks (Suh et al., 2022), we conjecture this method would struggle in the contact-rich ROBOMIMIC tasks.

## G. Understanding Exploration Behavior of **OGPO**

This section elaborates on the exploration dynamics of **OGPO** discussed in Section Section 4.2. We provide visualizations that clarify how **OGPO** expands the action manifold of pretrained policy distributions while maintaining stable policy improvement.

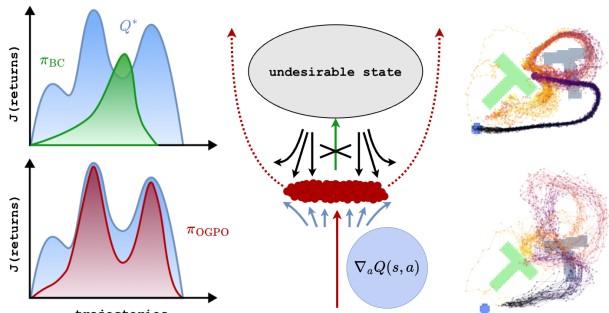
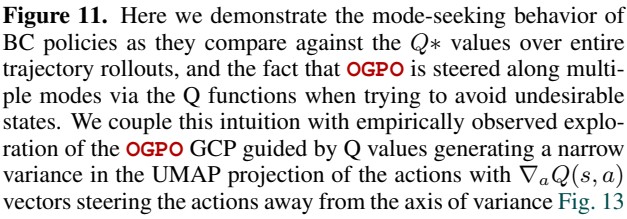
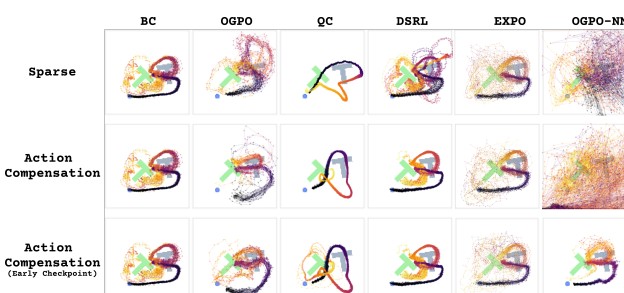

**Figure 11.** Here we demonstrate the mode-seeking behavior of BC policies as they compare against the $Q*$ values over entire trajectory rollouts, and the fact that **OGPO** is steered along multiple modes via the Q functions when trying to avoid undesirable states. We couple this intuition with empirically observed exploration of the **OGPO** GCP guided by Q values generating a narrow variance in the UMAP projection of the actions with $\nabla_a Q(s, a)$ vectors steering the actions away from the axis of variance Fig. 13

**Figure 12.** **OGPO** exhibits manifold expansion by making the policy more multi-modal as well as execution efficient. We show this by comparing policies learned by all algorithms with (top) Sparse reward setting, (middle) Sparse reward with $\Delta a_t$ compensation, and (bottom) Early stage policy Sparse reward with $\Delta a_t$ compensation

**Sample Efficiency vs. Execution Efficiency** In the training dynamics of **OGPO**, we observe two colliding optimization objectives: (1) **Sample Efficiency**: Minimizing the number of environment interactions required for policy convergence, and (2) **Execution Efficiency**: Minimizing the number of timesteps the policy takes to complete a task during inference.

**OGPO** excels at the former via off-policy stitching, but the latter introduces unique instabilities. The discount factor $\gamma < 1$ in the Bellman equation $Q(s, a) = r + \gamma Q_{\text{targ}}(s', a')$ creates a contraction map that conditions the policy to solve tasks as quickly as possible to maximize the expected return-to-go. This causes the GCP to generate actions that could potentially maximize the speed of achieving the goal, but do not necessarily abide by physical constraints like gravity, acceleration, and robot joint position and velocity limits. This explains the oscillations in the success rate during RL-finetuning induced by rapid policy convergence via Q functions.

**Visualizing Action Distributions.**    Figure 13 displays UMAP embeddings of action distributions at critical states from TOOLHANG, sampled at four stages: (i) end of BC pretraining, (ii) initial RL, (iii) mid RL, and (iv) end RL. We further run the natural baselines on the PushT task and rollout 50 trajectories and a no-negative grad ablation of **OGPO**. We show these rollouts in Figure 12 where the dark points represent the initial actions in the trajectory thorough the bright yellow ones, that represent the end of the trajectory. The classical Push-T sparse rewards allow **OGPO** to exploit the TD learning objective to learn executionally efficient policies with minimal corrective actions (denoted by end-stage yellow-orange points near the Push-T handle).

1. At the end of BC, all methods produce similar action distributions, confirming that differences emerge during RL finetuning rather than from initialization.

2. **OGPO** demonstrates the multi-modal conditioning of the policy via Q-functions which can be intuitively demonstrated in Fig. 11. To test this hypothesis, we add an action compensation term $r - \Delta a_t$

3. Methods like **QC** and **DSRL** show limited manifold expansion, remaining closer to the BC initialization. **EXPO**'s residual policy facilitates support expansion but not optimal policy extraction. This can be seed by a range of corrective actions being taken near the T-shape handle.

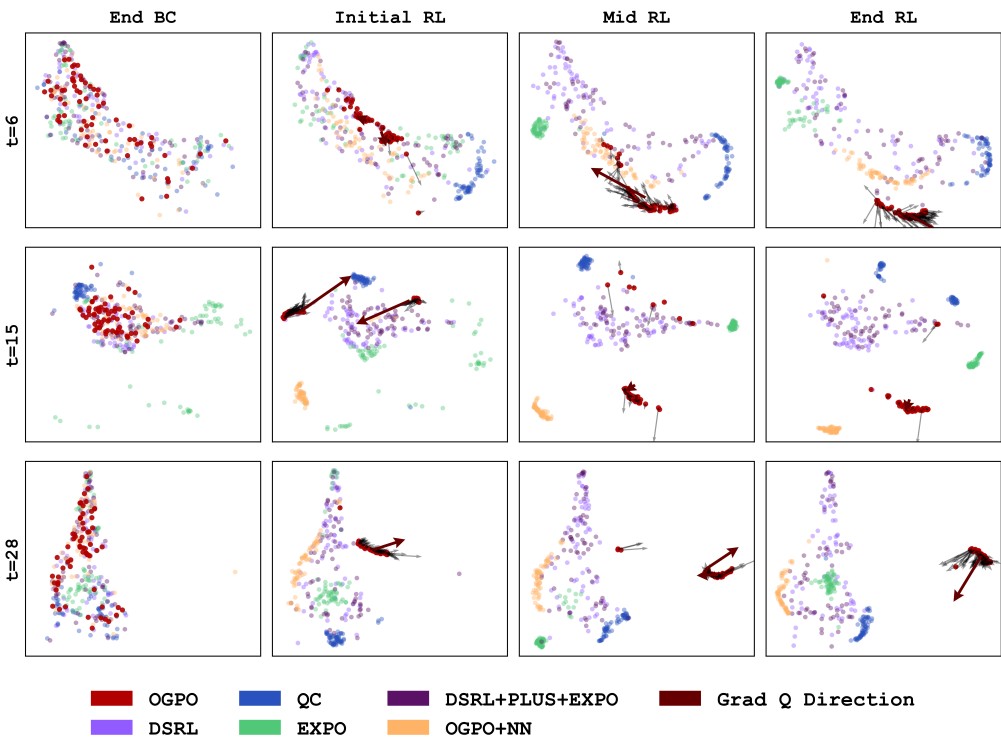

**Figure 13.** We consider 3 critical states (rows) in TOOLHANG shown in Fig. 14, and generate a UMAP plot of 64 action samples from **OGPO** and all the baselines combined and plotted separately at the end of BC pretraining, Initial-stage RL checkpoint Mid-stage RL checkpoint, and End-stage RL checkpoint for each critical state. We further show $\frac{d\text{UMAP}}{da}\nabla_a Q(s, a)$ vectors at each **OGPO** action to demonstrate the orthogonal conditioning of $\nabla_a Q(s, a)$ to the action spread at critical states.

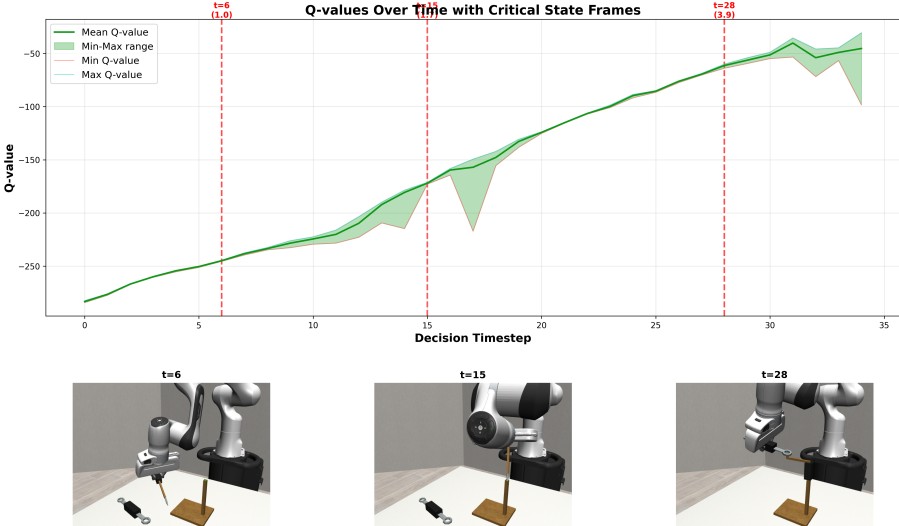

**Figure 14.** We rollout a successful **OGPO+** policy and obtain critical state frames in TOOLHANG and visualize the corresponding variance in $Q*$-values during the rollout

# H. Ablations and Limitations of **OGPO/OGPO+**

## H.1. **OGPO+** subcomponent ablations

We ablate the subcomponents of **OGPO+** to isolate where the maximal benefits come from. Following Figure 15, we conclude that success buffer significantly improves **OGPO**'s performance, whereas Best-of-$N$ sampling alone is not sufficient for policy improvement. This is consistent with prior work suggesting that Best-of-$N$ is a suitable verifier of critic training, and coupled with the success buffer SFT loss, improves **OGPO**'s sample efficiency.

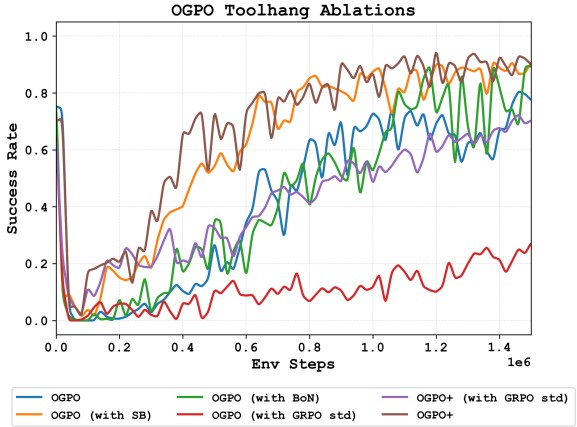

**Figure 15. OGPO** - **OGPO+** design ablations show that success buffer plays a crucial role in **OGPO+**'s performance.

## H.2. **OGPO** vs. **OGPO** with no-negative gradients

Removing negative gradients has a minimal impact on tasks like ROBOMIMIC SQUARE and TOOLHANG, where merely imitating high-valued actions is sufficient to sharpen policy distributions. However, for a task like TRANSPORT, where avoiding suboptimal policy modes is critical for task success, we observe worse performance for **OGPO−NN**. This suggests that negative advantages are important for mitigating suboptimal action distributions learned during pretraining.

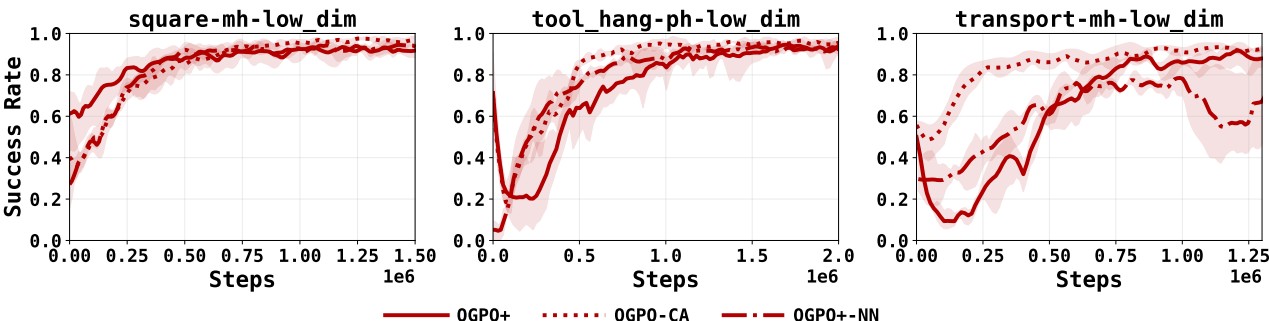

Figure 16. **OGPO+** vs **OGPO+** with no-negative gradients

## H.3. **BPTT** vs **OGPO**

The most direct way to train off-policy RL policies is to perform gradient ascent on the Q-values. Although this works for simpler policy parameterizations like Gaussian (Fujimoto et al., 2018), or Squashed Gaussian (Haarnoja et al., 2018) policies, directly using Q values to sequentially backpropagate through the GCP (also referred to as *Back Propagation Through Time (BPTT)*) can be unstable (Bengio et al., 1994). **OGPO** modifies the off-policy learning paradigm for a general class of GCPs by (1) retaining the TD error loss for Q function updates, and (2) using Q functions as substitutes for Monte Carlo rollouts and computing relative advantages $\hat{A}^G$ for PPO-style updates over the entire GCP chain for the policy updates.

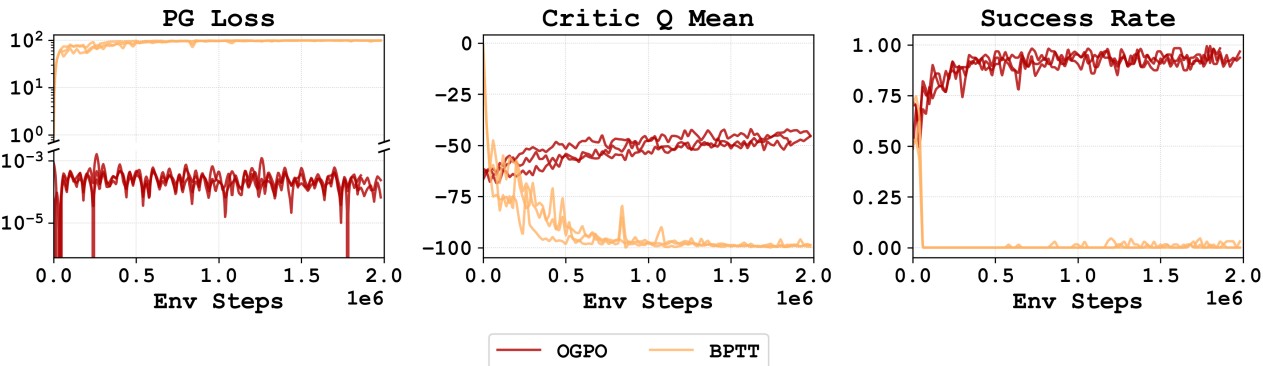

Figure 17. **BPTT** uses Q-values directly to backpropagate gradients along the entire GCP chain. This results in unstable gradients and poor convergence. In contrast, **OGPO** uses PPO-style policy gradient loss using Q-functions described Eq. (3.2). This results in stable gradients and sample-efficient convergence.

## H.4. **OGPO** v/s **OGPO+**, with and without GRPO std ($\sigma$)

GRPO formulation uses group relative advantage computation similar to **OGPO**. However, the GRPO advantage uses the standard deviation of the critic ensembles to normalize the advantage values. We found this to be empirically detrimental to **OGPO**'s success. We attribute this pattern to the sensitivity of the Annealed Importance Sampling ratio $\omega$ to very large and very small advantage values. We leave an extensive empirical validation of this sensitivity as future work.

## H.5. **OGPO** vs Steering + Residual Ablation

**OGPO** outperforms a custom Steering + Residual baseline with no offline data in the replay buffer.

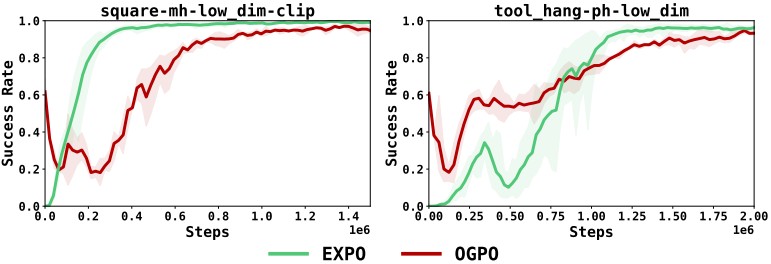

**Figure 18.** **OGPO+** comparison with an ablation of simultaneous steering and residual learning baseline: **S/R**

**EXPO** outperforms **OGPO** in task setting where offline_data = 0.5. Suggesting that for practitioners who can retain offline datasets during RL finetuning, residual corrections might work better than full policy finetuning.

**Figure 19.** Comparison to EXPO with Offline-Ratio = 0.5

## H.6. **OGPO+CA** Aligns Q-values Faster to ground truth MC-returns

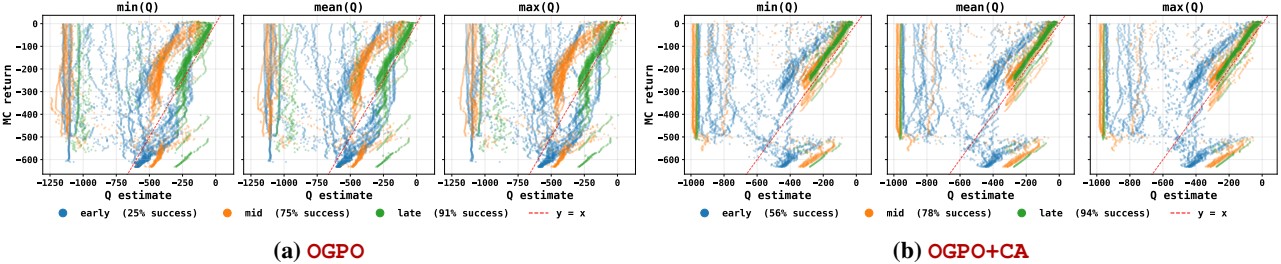

**Figure 20.** We take early-, mid-, and late- training checkpoints for **OGPO** and **OGPO+CA** to rollout 32 trajectories and visualize the min, mean, and max Q vs ground-truth, Monte-Carlo returns. (a) Shows **OGPO**'s Q values fluctuating widely between over- and under-estimating returns. (b) Shows **OGPO+CA**'s Q values converging more stably around the $y = x$ axis, demonstrating Q values accurately estimating returns.

# I. Related Work

We situate our work within the landscape of generative control policies, reinforcement learning for robotic control, and finetuning strategies for iterative generative models.

## I.1. Generative Control Policies

The success of diffusion models in image generation (Ho et al., 2020; Song et al., 2020; Rombach et al., 2022) has inspired their adoption for robotic control. Diffusion Policy (Chi et al., 2023) demonstrated that denoising diffusion probabilistic models (DDPMs) can effectively parameterize visuomotor policies by iteratively denoising action sequences conditioned on observations. Flow-matching policies (Lipman et al., 2022; Liu et al., 2022) offer a more efficient alternative by learning velocity fields that transport noise to action distributions through ordinary differential equations (ODEs), achieving comparable performance with fewer integration steps.

Recent work has sought to improve the generative modeling capacity. Notably, shortcut models (Frans et al., 2024) condition on desired step sizes to enable few-step generation, while consistency models (Song et al., 2023) distill multi-step diffusion into single-step generation. Recently, (Pan et al., 2025) introduced Minimally Iterative Policies (MIP), demonstrating that two-step regression-based policies can match full flow model performance, suggesting that distributional learning may be less critical than previously believed. Orthogonally, tokenized autoregressive policies such as FAST (Pertsch et al., 2025) encode continuous action chunks via discrete cosine transforms to enable efficient training of vision-language-action (VLA) models on high-frequency control data.

For **OGPO**, we demonstrate flow and diffusion-based policies as representative of the general IGP formulation and leave generalization to other formulations as future work.

### I.2. Reinforcement Learning for Robotic Policy Finetuning

The incorporation of Reinforcement Learning (RL) into robotic policy training mirrors the post-training paradigm in large language models (Ouyang et al., 2022; Shao et al., 2024). On-policy methods such as REINFORCE (Williams, 1992) and PPO (Schulman et al., 2017) update policies using only data from the current policy iteration, ensuring stable but sample-inefficient learning. DPPO (Ren et al., 2024) extends PPO to diffusion policies by computing policy gradients through the denoising chain, while Reinflow (Zhang et al., 2025) applies similar principles to flow-matching policies.

Off-policy algorithms promise greater sample efficiency by maintaining replay buffers of past experiences. Classical approaches such as SAC (Haarnoja et al., 2018), TD3 (Fujimoto et al., 2018), and REDQ (Chen et al., 2021) learn Q-functions from off-policy data to guide policy updates. Temporal difference learning mitigates the requirement of the policy to compute Monte Carlo return to the go. However, naive application to IGPs in the RL-finetuning regime can exhibit training instabilities due to large initial distributional shifts and value overestimation. To mitigate these, (Mark et al., 2024; Li et al., 2025) proposed using Q functions merely to rank stochastic policy actions and fine-tuning the policy using the Best-of-N actions. However, driving policy improvement via Q-function ranking can be inefficient as it requires exploration away from the mean values of the flow policy.

Concurrently, RL-100 (Lei et al., 2025) presents a comprehensive real-world RL framework built on diffusion policies, demonstrating deployment-grade success rates across eight manipulation tasks. RL-100 adopts the same bi-level MDP formulation and clipped PPO surrogate as DPPO, unifying imitation and reinforcement learning under a single objective across both offline and online stages, and additionally incorporates consistency distillation for high-frequency deployment. While RL-100 demonstrates impressive real-world reliability, its policy optimization remains fully on-policy, requiring iterative offline data expansion to achieve sample efficiency. **OGPO** instead decouples the bi-level MDP via off-policy critic learning, achieving comparable or superior sample efficiency in simulation without requiring multiple rounds of offline RL pre-training.

### I.3. Finetuning Strategies for Generative Control Policies

Existing approaches to finetuning GCPs differ along the axis of *what* is optimized. Steering methods, exemplified by **DSRL** (Wagenmaker et al., 2025), optimize the distribution over initial noise $a_K$ while freezing the pretrained denoising network. This constrains policy improvement within the support of the pretrained IGP distribution. Residual policy approaches such as **EXPO** (Dong et al., 2025) train an additional network $\pi^{\text{res}}$ that modifies the final action $a_{\text{res}} = \pi^{\text{res}}(a_{t,0}, s_t)$, allowing mode shifts within the BC policy support but fails to facilitate discovery of new behaviors.

Policy-agnostic RL (PA-RL) (Mark et al., 2024) and Q-chunking (**QC**) (Li et al., 2025) employ Q-functions to rank behavior cloned policies with high-value actions or use $\nabla_a Q(s, a)$. In the image generation domain, Flow-GRPO (Liu et al., 2025) concurrently applied GRPO (Shao et al., 2024) to flow matching models for text-to-image alignment, sharing with **OGPO** the ODE-to-SDE conversion for injecting stochasticity into deterministic flow policies and the use of group-relative advantage estimation over parallel denoising trajectories. However, Flow-GRPO operates in the on-policy, bandit-like setting: rewards are terminal (image-level), the "environment" is a single-step generation with no dynamics, and advantages are estimated via group normalization of final rewards rather than learned Q-functions.

In contrast, **OGPO** addresses the multi-step robotic control setting, where off-policy TD-learning is essential for sample efficiency across long environment horizons, and the two-level MDP structure enables reuse of costly environment transitions while performing on-policy updates purely within the denoising MDP. However, in addition to zero-order optimization via Q functions, **OGPO** performs SFT via Success Buffer actions for enhanced sample efficiency.

# J. Environment Details

## J.1. FRANKA-KITCHEN

The FRANKA-KITCHENbenchmark (Gupta et al., 2019) tests multi-task sequential manipulation with compositional task structure. The environment features a 9-DoF Franka robot that must manipulate 4 kitchen objects (microwave, kettle, light switch, slide cabinet) to desired goal configurations in a specific sequence. This environment is particularly challenging due to its requirement for long-horizon planning and the need to compose multiple subtasks correctly.

**State and Action Spaces:** The state space consists of robot joint positions, joint velocities, and object states (state_dim = 60). Actions are 9-dimensional continuous controls for the robot joints (action_dim = 9), normalized to $[-1, 1]$.

**Task Horizon and Other Parameters:** FRANKA-KITCHENtasks have a medium horizon of approximately 280 timesteps. We use $\gamma = 0.99$ to account for the medium-length temporal dependencies across subtasks. The action chunk size is set to $h = 4$ to provide temporal smoothness while maintaining reactivity.

**Datasets:** We use three offline datasets from D4RL (Fu et al., 2020):

- KITCHEN-COMPLETE: Complete demonstrations of all 4 subtasks in the correct sequence

- KITCHEN-MIXED: Randomized subtask orders where the desired sequence is not completed sequentially

- KITCHEN-PARTIAL: Partial subtrajectories of the desired task

**Reward Structure:** We use a sparse reward structure with a base reward of -7. Each successful subtask completion adds +1, with the final subtask providing +3 upon success. This yields a maximum reward of 0 for completing all subtasks.

## J.2. Robomimic

The ROBOMIMIC benchmark (Mandlekar et al., 2021) provides high-precision manipulation tasks that test fine-grained control and multi-step reasoning. We evaluate on three of the most challenging tasks that represent different aspects of real-world manipulation:

**Square (SQUARE):** A medium-horizon fine-grained insertion task requiring precise alignment and insertion of a square peg. This task tests contact-rich manipulation with tight tolerances.

- state_dim: 14 (robot end-effector pose, object pose)

- action_dim: 7 (6D end-effector control + gripper)

- Horizon: 400 timesteps

- $\gamma = 0.99$

- Action chunk size: $h = 4$

- Dataset: Multi-Human (MH) mixed proficiency

**Tool Hang (TOOLHANG):** A long-horizon, highly-precise multi-step insertion task requiring the robot to grasp a tool and hang it on a rack. This task demands both coarse positioning and fine-grained alignment across multiple phases.

- state_dim: 14

- action_dim: 7

- Horizon: 1000 timesteps

- $\gamma = 0.999$ (higher due to longer horizon)

- Action chunk size: $h = 8$ (larger chunks for smoother long-horizon execution)

- Dataset: Proficient-Human (PH), BC stopped at 50% success rate

**Transport (TRANSPORT):** A bi-manual, multi-step, long-horizon object transfer task where two robot arms must coordinate to transport an object. This tests both individual arm control and bi-manual coordination.

- `state_dim`: 28 (dual arm configuration)
- `action_dim`: 14 (7 per arm)
- Horizon: 800 timesteps
- $\gamma = 0.999$ (higher due to longer horizon)
- Action chunk size: $h = 8$
- Dataset: Multi-Human (MH) mixed proficiency

**Reward Structure:** All Robomimic tasks use sparse rewards: -1 for each non-successful step, with the final successful step returning 0.

**Note on Hyperparameters:** The different gamma values reflect the relationship between discount factor and task horizon. Longer horizon tasks (TOOLHANG, TRANSPORT) require larger gamma (0.995) to properly credit distant actions, while medium-horizon tasks (SQUARE) use smaller gamma (0.99). Similarly, longer tasks benefit from larger action chunks ($h = 8$) for smoother execution. Importantly, both gamma and chunk size are independent of action dimensionality.

### J.3. Adroit Hand

The Adroit Hand benchmark tests dexterous manipulation with a 24-DoF anthropomorphic robotic hand performing high-precision, contact-rich tasks. This environment is particularly challenging due to the high-dimensional action space, under-actuated dynamics, and the need for coordinated finger movements.

We evaluate on four standard tasks:

- `AdroitHandDoor-v1`: Door opening requiring articulated finger coordination to grasp and turn a handle
- `AdroitHandHammer-v1`: Hammering a nail with precise force control and wrist articulation
- `AdroitHandPen-v1`: In-hand pen reorientation requiring complex finger gaiting
- `AdroitHandRelocate-v1`: Object relocation requiring coordinated grasping and translation

**State and Action Spaces:**

- `state_dim`: 45 (24 joint positions + 24 joint velocities + object state)
- `action_dim`: 24 (continuous control for each DoF)
- Actions normalized to $[-1, 1]$

**Task Horizon and Temporal Parameters:**

- Horizon: 200 timesteps (medium-horizon tasks)
- $\gamma = 0.95$
- Action chunk size: $h = 4$ for stabilized policy execution

**Datasets:** We use expert demonstration datasets provided via the D4RL/Minari interface for pretraining the base policy.

**Evaluation:** Following prior work, we evaluate performance using the normalized return provided by the environment, scaled to $[0, 100]$.

## J.4. LIBERO

The LIBERO benchmark (Liu et al., 2023) tests vision-based, language-conditioned manipulation for multi-task learning and generalization. Unlike the previous environments, which use state-based observations, LIBERO provides pixel observations and requires following natural-language instructions, thereby testing both visual understanding and instruction-following capabilities.

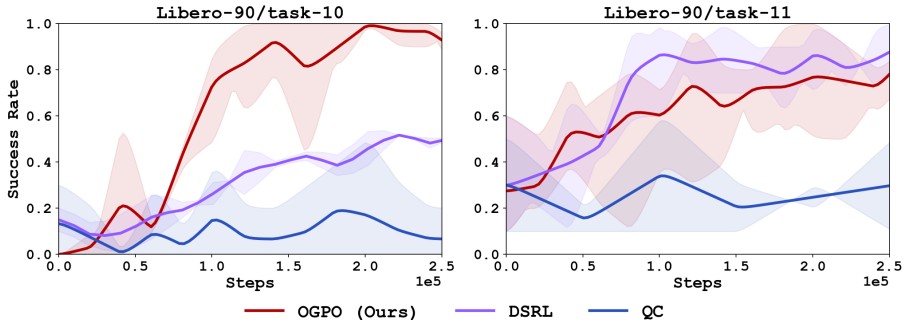

**Figure 21.** We compare **OGPO** with **DSRL** and **QC** on pixel-based observations and natural language guidance tasks from the LIBERO benchmark

**Observation and Action Spaces:**

- Observations: RGB images (128×128×3 pixels)

- `action_dim`: 7 (6D end-effector control + gripper)

- Actions normalized to $[-1, 1]$

**Task Structure:** LIBERO features procedurally generated tasks with natural language instructions. Tasks require understanding spatial relationships and object attributes from both visual and linguistic modalities.

**Reward Structure:** All Libero tasks use sparse rewards: -1 for each non-successful step, with the final successful step returning 0.

**Task Horizon and Temporal Parameters:**

- Horizon: 1000 timesteps (long-horizon tasks)

- $\gamma = 0.999$ (for **OGPO**, **DSRL**), 0.99 (for **QC** since we found this leads to better performance)

- Action chunk size: $h = 8$

**Training and Evaluation Setup:** The base policy is trained on demonstrations from 10 tasks (`task_id` $\in \{0, \ldots, 9\}$) in the Libero-90 dataset and evaluated on 2 unseen downstream tasks (`task_id` $\in 10, 11$) to test generalization capabilities. This setup explicitly tests the ability to transfer learned manipulation skills to novel task descriptions and object configurations. Since LIBERO is a language-conditioned benchmark, for both the actor and critic, we follow a widely used design from prior work (Walke et al., 2023; Nakamoto et al., 2024a): language instructions are first processed by a frozen MUSE encoder (Yang et al., 2019) and then passed to an IMPALA encoder (Espeholt et al., 2018) with FiLM conditioning (Perez et al., 2018).

# K. Hyper-parameters and Initialization

## K.1. Initialization and Warm Starting

**OGPO** accommodates two primary settings based on data availability, each with corresponding algorithmic choices for initialization. **Setting 1: Offline data available.** When an offline dataset $\mathcal{D}_{\text{off}}$ is available, we pre-train our policy $\bar{\pi}_\theta^{\text{BC}}$

on $\mathcal{D}_{\text{off}}$ using the appropriate BC loss. The `use_offline` flag is toggled `True`, enabling offline data sampling reuse determined by the ratio $r_{\text{offline}}$.

**Setting 2: No offline data (online-only).** We finetune a pre-trained IGP with *no* additional demonstration data which has some small but non-trivial base success rate (>10%). The `use_offline` flag is toggled `False`.

In both settings, the online replay buffer $\mathcal{D}_{\text{roll}}$ is initialized with $N_{\text{warmup}}$ $\bar{\pi}_\theta$ rollouts, where $\bar{\pi}_\theta \leftarrow \bar{\pi}_\theta^{\text{BC}}$. Finally, we initialize an ensemble of Q-functions $Q_{\phi_{1,\dots,M}}$ with random weights and, importantly, find that no offline RL pretraining yields the highest sample efficiency. We defer the details of the offline RL ablations to Algorithm 3.

### K.2. Hyperparameters

In this section, we list all the hyper parameters we use for **OGPO** across different benchmarks. Table 2 shows the maximum episode lengths we use for each environment.

| Environment | Max Episode Length |
|---|---|
| square | 400 |
| transport | 800 |
| tool_hang | 1000 |
| kitchen (all) | 600 |
| adroit (all) | 200 |

**Table 2.** Environment maximum episode lengths

We first list the common **OGPO** hyper parameters. Unless otherwise stated, these remain constant throughout all our experiments. These are in Table 3.

| Parameter | Default Value |
|---|---|
| lr | 3e−4 |
| actor_lr | 3e−4 |
| critic_lr | 3e−4 |
| ppo_lr | 4.5e−5 |
| tau | 0.05 |
| actor_tau | 0.05 |
| discount | 0.99 |
| batch_size | 256 |
| ppo_batch_size | 256 |
| actor_hidden_dims | (512, 512, 512, 512) |
| value_hidden_dims | (512, 512, 512, 512) |
| num_qs | 10 |
| q_agg | mean |
| subsample_bon | True |
| flow_steps | 10 |
| grpo_num_samples | 32 |
| clip_epsilon | 0.01 |
| entropy_coeff | 0.0 |
| bc_coeff | 1.0 |
| constant_noise_std | 0.01 |
| actor_scheduler | cosine |
| critic_scheduler | constant |
| actor_warmup_steps | 2000 |
| actor_decay_steps | 50000 |
| actor_end_value | 2e−5 |
| critic_warmup_steps | 500 |
| critic_decay_steps | 5000 |
| critic_end_value | 0.0 |
| actor_weight_decay | 0.0 |
| critic_weight_decay | 1e−5 |
| horizon_length | 4 |
| policy_type | flow |

**Table 3.** OGPO agent default hyperparameters.

In Table 4, we list down all ROBOMIMIC specific hyper-parameters that are used for our experiments.

| Hyperparameter | SQUARE | TOOLHANG | TRANSPORT |
|---|---|---|---|
| **Training Steps** | | | |
| offline_steps | 500,000 | 500,000 | 1,000,000 |
| online_steps | 2,000,000 | 3,000,000 | 6,000,000 |
| start_training | 20,000 | 25,000 | 40,000 |
| **RL Hyperparameters** | | | |
| horizon_length | 4 | 8 | 8 |
| discount | 0.99 | 0.999 | 0.999 |
| tau | 0.05 | 0.05 | 0.05 |
| utd_warmup | 1 | 1 | 1 |
| utd_online | 1 | 1 | 1 |
| **Q-Network** | | | |
| num_qs | 10 | 10 | 10 |
| q_agg | mean | mean | mean |
| subsample_bon | True | True | True |
| best_of_n | 8 | 8 | 8 |
| value_hidden_dims | (512,512,512,512) | (512,512,512,512) | (512,512,512,512,512) |
| **BC Regularization** | | | |
| use_bc_regularization | True | True | True |
| bc_coeff | 1.0 | 1.0 | 1.0 |
| pg_coeff | 1.0 | 1.0 | 1.0 |
| clip_bc (atmost 50% success rate) | True | True | False |

**Table 4.** OGPO hyperparameters for Robomimic environments.

In Table 5, we list all hyper parameters we use for the various FRANKA-KITCHEN environments.

| Hyperparameter | KITCHEN-COMPLETE | KITCHEN-MIXED | KITCHEN-PARTIAL |
|---|---|---|---|
| **Training Steps** | | | |
| offline_steps | 1,000,000 | 1,000,000 | 1,000,000 |
| online_steps | 3,000,000 | 3,000,000 | 3,000,000 |
| **RL Hyperparameters** | | | |
| horizon_length | 4 | 4 | 4 |
| discount | 0.99 | 0.99 | 0.99 |
| tau | 0.05 | 0.05 | 0.05 |
| utd_warmup | 1 | 1 | 1 |
| utd_online | 1 | 1 | 1 |
| **Q-Network** | | | |
| num_qs | 10 | 10 | 10 |
| q_agg | mean | mean | mean |
| subsample_bon | True | True | True |
| best_of_n | 8 | 8 | 8 |
| **BC Regularization** | | | |
| use_bc_regularization | True | True | True |
| bc_coeff | 0.1 | 0.1 | 0.1 |
| clip_bc | False | False | False |

**Table 5.** OGPO hyperparameters for FRANKA-KITCHEN

In Table 6, we list all hyper parameters we use for the various FRANKA-KITCHEN environments.

| Hyperparameter | Door-v1 | Pen-v1 | Hammer-v1 | Relocate-v1 |
|---|---|---|---|---|
| **Training Steps** | | | | |
| offline_steps | 50,000 | 50,000 | 50,000 | 50,000 |
| online_steps | 500,000 | 500,000 | 500,000 | 500,000 |
| **RL Hyperparameters** | | | | |
| horizon_length | 4 | 4 | 4 | 4 |
| discount | 0.95 | 0.95 | 0.95 | 0.95 |
| tau | 0.05 | 0.05 | 0.05 | 0.05 |
| utd_warmup | 1 | 1 | 1 | 1 |
| utd_online | 4 | 4 | 4 | 4 |
| **Q-Network** | | | | |
| num_qs | 10 | 10 | 10 | 10 |
| q_agg | min | min | min | min |
| subsample_bon | False | False | False | False |
| best_of_n | 8 | 8 | 8 | 8 |
| **BC Regularization** | | | | |
| use_bc_regularization | True | True | True | True |
| bc_coeff | 1.0 | 1.0 | 1.0 | 1.0 |
| clip_bc | True | True | True | True |

**Table 6.** OGPO hyperparameters for Adroit.

In Table 7, we list all hyperparameters we use for the Libero environments.

| Hyperparameter | Libero |
|---|---|
| **Training** | |
| offline_steps | 50,000 |
| online_steps | 250,000 |
| actor_tau | 0.001 |
| batch_size | 64 |
| constant_noise_std | 0.01 |
| grpo_num_samples | 8 |
| **RL Hyperparameters** | |
| horizon_length | 8 |
| discount | 0.999 |
| tau | 0.05 |
| utd_online | 1 |
| **Q-Network** | |
| num_qs | 10 |
| q_agg | mean |
| encoder | impala_small |
| value_hidden_dims | (128, 128, 128) |
| **BC Regularization** | |
| use_bc_regularization | False |
| offline_ratio | 0 |

**Table 7.** OGPO hyperparameters for Libero.

