# OpenReview forum: "OGPO: Sample Efficient Full-Finetuning of Generative Control Policies"
_ICML.cc/2026/Conference — ICML 2026 regular_

### Official Review · Reviewer_7F7d · 2026-03-12

**Soundness:** 3
**Presentation:** 2
**Significance:** 2
**Originality:** 3
**Overall Recommendation:** 4
**Confidence:** 3

**Summary:**

Regarding the issue of how generative control strategies can be efficiently optimized through reinforcement learning, this paper views the optimization of generative control strategies as a bi-level MDP problem. It proposes the Off-policy Generative Policy Optimization (OGPO) algorithm to maximize the data reuse rate. By modifying the PPO objective mechanism, it propagates the policy gradients throughout the complete generation process of the policy.

**Compliance With Llm Reviewing Policy:**

Affirmed.

**Final Justification:**

The authors provided a thorough rebuttal that successfully addressed my primary concerns regarding the soundness and empirical evaluation of the paper.

**Key Questions For Authors:**

There are many typo errors in the paper, such as:
1. In Line 187，state and action at step $k$ are $a_{t,k}$ and $a_{t,k−1}$, respectively.
2. In Line 284, Finally, we describe limitations of our approach in

**Limitations:**

Yes.

**Strengths And Weaknesses:**

Strengths:
1. The paper proposes a novel approach to address the gradient propagation issue when using PPO for optimization in generative control policies.
2. Sufficient experiments are provided to verify the effectiveness of the method, including four environments.

Weaknesses:
1. The article is a bit rough, with some typo errors and issues regarding writing norms.
2. In Eq 3.2, should $a_{t,0}$ in $\hat{A}=Q_{targ}(s_t,a_{t,0})-\hat{V}$ be changed to $a_{t,k}$? I want to know how the Q-network is trained, because in denoising MDP, the reward at each moment cannot be known.
3. In Figure 3, OGPO does not show a significant advantage in the Adroit environment. Moreover, when incorporating the same optimizations from OGPO, the performance of other baselines did not improve. Does this indicate that the method lacks generalization?
4. It would be better to place the ablation experiments in the main text. In Figure 15, it would be best to explain the meanings of SB, BoN, and GRPO std in the caption. With GRPO std, the performance of OGPO did not improve. Additionally, does the figure present the ablation of behavior cloning regularization? If it shows the difference between OGPO (with SB) and OGPO+, then Behavior cloning regularization might be an ineffective component.
5. It would be great if some real-world experiments could be provided, but it's fine if not.

---

> ### Author Rebuttal · Authors · 2026-03-31
>
> We thank Reviewer 7F7d for engaging with our paper. We appreciate the recognition of the novelty of our gradient propagation approach and the acknowledgment that we provide sufficient experiments across four environments. We address each concern below.
>
> **W1: The article is a bit rough, with typo errors and writing issues**
>
> We address all of the points below. If there are other specific writing issues that the reviewer could point out after the following clarifications, we would be happy to address them.
>
> **W2: In Eq 3.2, should $a_{t,0}$ be changed to $a_{t,k}$? How is the Q-network trained in the denoising MDP?**
>
> We clarify that $a_{t,0}$ is correct. The Q-networks are trained on the environment MDP, not the denoising MDP. They take as input the state $s_t$ and the fully denoised action $a_{t,0}$. As the reviewer correctly notes, Q-values at intermediate denoising steps cannot be meaningfully defined because intermediate noise samples are not environment actions. This can be viewed as having a terminal reward at the denoising step k=0 (denoted by the red arrow in Fig. 1).
>
> In order to retain the sample-efficiency benefits of Q functions, OGPO decouples the off-policy updates with the Q functions only learning expected returns over $a_{t,0}$, and the on-policy gradient updates that compute the importance sampling term over the entire $a_{t,k}$ chain and weigh the log probabilities using the advantages estimated by the group-relative Q functions over the final denoising MDP steps.
>
> Section 3, Algorithm 1, and the Pseudocode in Appendix Sec 1 give a detailed description of the same. We would love your input to improve the communication of our method for the final manuscript.
>
> **W3: OGPO does not show a significant advantage in Adroit.**
>
> Adroit tasks can be solved by standard SAC, as these environments provide dense rewards. They do not require the expressiveness of generative policies, which play a crucial role in sparse-reward environments. OGPO’s demonstrates the most significant performance gains in environments that have been historically difficult for sample-efficient GCP fine-tuning: Transport and ToolHang (800-step horizons with sparse rewards), where we show clear advantages over off-policy GCP baselines.
>
> **W4: Moreover, the same optimizations from OGPO did not improve other baselines. Does this indicate lack of generalization?**
>
> Regarding the OGPO+ optimizations not transferring to other baselines: We do not claim these are general-purpose improvements. The success buffer and Best-of-N selection are synergistically aligned with OGPO’s full-policy fine-tuning mechanics. Note, Best-of-N was inspired by the Q-chunking baseline. We added Success Buffer as an additional feature to stabilize our policy updates, and demonstrated a naive application to the baselines for a fair comparison. Understanding why these interventions fail for steering and residual-based methods warrants future study.
>
> **W5: It would be better to place the ablation experiments in the main text**
>
> We agree. Due to space constraints, full ablations are in Appendix Sec. J with a summary in Section 4.3. With the additional permitted page, we will expand the ablation discussion in the main body. We welcome guidance on which specific ablations to prioritize.
>
> **W6: Real-world experiments would be valuable**
>
> We agree that real-world validation would strengthen the contribution. We are currently extending our image-based experiments with VLM backbones taken from VLAs like the open-sourced $\pi_{0.5}$, with the aim of deploying on hardware. We view this as an important next step and will report results in the final manuscript.
>
> **Typo errors in the paper**
> Fixed, thank you!
>
> We sincerely appreciate Reviewer 7F7d’s time in reviewing our work. We would genuinely welcome any additional specific feedback—particularly on the technical aspects of the method or the experimental evaluation—as this would help us further improve the manuscript.

---

> > ### Author Rebuttal · Reviewer_7F7d · 2026-04-03
> >
> > Given that my main concerns have been resolved, I am raising my score to 4.

---

### Official Review · Reviewer_7Knu · 2026-03-13

**Soundness:** 3
**Presentation:** 3
**Significance:** 3
**Originality:** 3
**Overall Recommendation:** 4
**Confidence:** 3

**Summary:**

This paper addresses the challenge of sample-efficiently fine-tuning Generative Control Policies (GCPs), such as diffusion and flow-based models, for robotic continuous control. The authors identify that partial fine-tuning (e.g., steering or residual learning) lacks expressive exploration, while full-trajectory on-policy fine-tuning (e.g., DPPO) suffers from extreme sample inefficiency in the environment. To resolve this, the authors propose Off-policy Generative Policy Optimization (OGPO), a decoupled architecture that uses off-policy Temporal Difference (TD) learning on the true environmental MDP to train a Critic, and applies on-policy PPO purely on the computational "denoising MDP" to update the Actor. An enhanced version, OGPO+, incorporates Best-of-N inference and a Success Buffer (behavior cloning regularization) to stabilize training. Results across Robomimic, Franka Kitchen, Adroit, and Libero demonstrate that OGPO achieves state-of-the-art sample efficiency and high success rates, uniquely recovering policies from poor behavior cloning (BC) initializations without requiring expert data in the online replay buffer.

**Compliance With Llm Reviewing Policy:**

Affirmed.

**Final Justification:**

The authors fully resolved my concerns, I would keep my rating.

**Key Questions For Authors:**

1. How does the immense computational overhead of generating multiple parallel denoising trajectories for the PPO baseline calculation affect per-iteration runtime and memory usage? Could you provide a wall-clock time comparison and FLOP estimates against DPPO and DSRL?
2. The online experiments use a base policy pre-trained to at least a 50% success rate. How does OGPO+ perform if the initial BC policy has a very low success rate (e.g., 5% or 10%)? Does the Success Buffer mechanism fail to bootstrap if it cannot collect successful trajectories early in training?
3. How do you reconcile the fundamental objective conflict in OGPO+? PPO attempts to expand the action manifold to maximize $R$, while the Success Buffer's $\mathcal{L}_{BC}$ strictly anchors the policy to historical data. Does this $\lambda_{BC}$ hyperparameter effectively limit the algorithm strictly to local mode-seeking rather than global exploration?
4. What is the clip ratio during training? From clipping to not clipping? This can prove the exploration capabilities of the method.

**Limitations:**

The authors have partially discussed limitations, explicitly acknowledging the potential for the PPO policy extraction to over-exploit the Q-function (leading to performance oscillations) and noting that the method is needlessly expressive and slow when large amounts of expert data are available. However, a dedicated limitations section must explicitly address: (1) the massive computational (GPU) bottleneck introduced by the parallel trajectory rollouts required for the actor update; (2) the Bootstrapping Paradox, acknowledging that the method requires a highly competent BC initialization to function in sparse reward environments; and (3) the fragility of the ODE-to-SDE noise correction mechanism, which forces the addition of an auxiliary noise-prediction head and may become unstable in extremely high-dimensional continuous action spaces.

**Strengths And Weaknesses:**

**Strengths:**

1. **Innovative Decoupled Architecture**: The conceptual separation of the physically expensive environmental MDP (solved via off-policy Critic) and the computationally cheap denoising MDP (solved via on-policy PPO) is an elegant and highly practical solution. It successfully bridges the gap between sample efficiency and stable generative policy extraction.
2. **Zero-Order Optimization over BPTT**: By utilizing the Critic as a terminal reward and optimizing the generative chain via PPO (using likelihood ratios), the algorithm cleverly sidesteps the gradient explosion and inaccuracies associated with Backpropagation Through Time (BPTT) and $\nabla_a Q(s,a)$ in contact-rich, non-smooth tasks.
3. **Strong Empirical Performance in Low-Data Regimes**: The ability to fully fine-tune GCPs to near-perfect success rates starting from a clipped BC checkpoint (50% success rate)—without needing to sample from the original offline dataset during online RL—is a significant milestone for realistic robot deployment scenarios.

**Weaknesses:**

1. **Over-reliance on the Success Buffer (Conflicting Objectives)**: The method relies heavily on the "Success Buffer" ($\mathcal{L}_{BC}$) in OGPO+ to prevent the PPO actor from collapsing due to reward hacking under sparse rewards. This imposes a double restriction (PPO clipping + BC) that mathematically contradicts PPO's objective to expand the action manifold. This suggests the off-policy Critic alone is fundamentally insufficient to safely guide the actor, limiting true global exploration.
2. **Hidden Computational Overhead**: While the paper emphasizes *sample* efficiency (few environment interactions), it completely masks the *computational* cost. Evaluating the PPO baseline $\hat{V}$ requires generating $N_{group}$ parallel, full-denoising trajectories per state for every update step (highly analogous to GRPO in LLMs). This necessitates massive GPU forward passes, making the algorithm's wall-clock time potentially prohibitive compared to residual or steering baselines.
3. **The Bootstrapping Paradox**: The algorithm relies on pre-training the base policy to a $\sim$50% success rate to warm up the online buffer with successful trajectories. If the initial BC policy has a very low success rate, the randomly initialized Critic will suffer from extreme sparse-reward bootstrapping failures, and the Success Buffer will remain empty.

---

> ### Author Rebuttal · Authors · 2026-03-31
>
> We thank Reviewer 7Knu for the detailed engagement with our work. We appreciate the recognition of the kind appraisal of the elegance and novelty of our method.  We address each concern below.
>
> **W1: Over-reliance on the Success Buffer (Conflicting Objectives)**
>
> Success buffer training does not present a conflicting objective with PPO: [1] propose a Conservative Policy Iteration (CPI) objective that was used by TRPO [2] to derive a KL divergence penalty in addition to the surrogate objective using importance sampling. Schulman et al. in PPO [3] claim that the surrogate objective, which uses the maximum KL divergence between the current and an old policy, forms a lower bound on the performance of the policy.  PPO [3] proposes clipping the probability ratios as a computationally efficient surrogate for this objective.
>
> As the flow-matching loss is an ELBO for KL between policy and action distributions [4], one can enforce additional KL regularization by imitating data in the pre-training buffer. This (1) it requires BC data in the buffer; (2) limits policy improvement *beyond* the imperfect BC policy. The success buffer remedies both by enforcing KL with respect to new, on-policy data, acting like a KL constraint, and filtering trajectories so only the successful are retained. This offers a third benefit of (3) mitigating the overloading between “fast completion” and “high success”  which are conflated in typical sparse reward settings (total return = -1*trajectory length) by encouraging learning from past successes. Hence, we posit that the success buffer both enforces intuitions from CPI/TRPO and combats reward misalignment by biasing towards successful policies. We will incorporate this discussion in the revision.
>
> [1] S. Kakade and J. Langford. “Approximately optimal approximate reinforcement learning”. 2022.
>
> [2] Schulman, John, et al. "Trust region policy optimization." 2015.
>
> [3] Schulman, John, et al. "Proximal policy optimization algorithms." 2017.
>
> [4] Ho, Jonathan, Ajay Jain, and Pieter Abbeel. "Denoising diffusion probabilistic models." 2020.
>
> **W2: Hidden Computational Overhead**
>
> This is a valid limitation which we will clarify in our “Limitations”. While steering / residual baselines run faster with simulations, they are bottlenecked by robot execution speeds in the real world. In many cases, e.g. in modern VLAs, the VLM backbone is more expensive than the action head. However, since we can cache the VLM backbones and optimize smaller action heads via OGPO, this results in a much lower overhead. Further, real experiments can allow asynchronous jax updates to enable faster runtime; we keep updates synchronous in simulation to aid reproducibility. We anticipate with these updates, OGPO updates will not be a bottleneck on hardware.
>
> **W3: The Bootstrapping Paradox**
>
> We stress that offline-to-online training is standard in this literature. Moreover, OGPO’s bootstrapping requirements are mild compared to other baselines. We pre-train to ~50% BC success rate and run warmup rollouts before RL begins. Even at a 10% initial success rate, 10 warmup rollouts are sufficient to populate the success buffer with at least one successful trajectory — the buffer does not remain empty. During the duration of this rebuttal, we ran experiments with a 10% success rate BC policy and found OGPO to converge within 1.5e6 steps compared to DPPO which converges in 2e7, and DSRL which does not converge at all. All of these results, will be included in the final manuscript.
>
> **Q1.1: Computational overhead, wall-clock time comparison:**
>
> Our implementation is written in Jax with JIT-compiled functions for the policy gradient and Q function updates. And although OGPO runs slower than DSRL and DPPO in wall clock time in a simulation environment, higher sample efficiency results in significantly lower wall clock times during real world RL by the optimizations described in W2.
>
> **Q1.2: FLOP estimates against DPPO and DSRL?** (In simulation)
>
> Time for 2 million steps: DPPO: 4.5 hrs, DSRL: 4.5 hrs, OGPO: 20 hrs
>
> Steps and Time for >=95% success: DPPO (8.5 million steps): 18.5 hrs, DSRL (4 million steps): 9 hrs, OGPO (750k steps): 10 hrs
>
> We will add wall-clock comparisons and per-step FLOP estimates to the revised manuscript.
>
> **Q2. How does OGPO+ perform with a very low initial BC success rate (e.g., 5% or 10%)?**
> Please refer to W3
>
> **Q3: How do you reconcile the fundamental objective conflict in OGPO+?**
> Please refer to the discussion in W1.
>
> **Q4: What is the clip ratio during training?**
> As mentioned in Appendix Sec. L, clip ratio denoted as \emph{clip_epsilon} for all benchmarks using OGPO is $0.01$.
>
> **Limitations:**
> (1) and (2) are addressed above. (3) The ODE-to-SDE correction is described in detail in Appendix Sec. H and empirically shown to stabilize OGPO at action dims 7 for square and tool_hang and 14 for transport in Fig. 9. Do you have feedback for us to improve the accessibility or delivery of the section?

---

> > ### Author Rebuttal · Reviewer_7Knu · 2026-04-07
> >
> > The authors fully resolved my concerns, I would keep my rating.

---

### Official Review · Reviewer_T34B · 2026-03-13

**Soundness:** 3
**Presentation:** 3
**Significance:** 2
**Originality:** 2
**Overall Recommendation:** 4
**Confidence:** 4

**Summary:**

OGPO aims to make the DDPO/ReinFlow template more sample-efficient with respect to environment MDP interactions. While DDPO/ReinFlow uses on-policy PPO over the full bi-level MDP (environment and inner denoising/flow steps), OGPO learns an off-policy critic on the environment MDP and uses the critic to formulate the terminal reward signal for on-policy PPO on the inner denoising MDP. The expensive interactions for robotics (environment MDP) are learned off-policy, while the cheap ones (denoising/flow MDP) are learned on-policy, making the entire pipeline aligned with cost-efficiency concerns.
OGPO requires an array of tricks/techniques:
- an ensemble of critics
- several denoising parallel tracks (forming a “group”) to amortize Q-estimate with 1+ proposal predicted action (chunks)
- a ODE-SDE correction proposed by Albergo et al. (2023) for the FM sampling
- a success replay buffer from Oh et al., (2018) to regularize the policy with a BC loss (OGPO+)

**Compliance With Llm Reviewing Policy:**

Affirmed.

**Final Justification:**

I will maintain my score.

**Key Questions For Authors:**

Questions:
- Why do the authors put so much emphasis on using zero-th order method, considering the entire operations the authors propose are independently encapsulated within the denoising process inbetween each step of the MDP? Is this just due to the authors finding that, empirically, "backpropagation through the denoising steps, as opposed PPO, often fails catastrophically”?
- It is concerning that the method is brittle wrt the trick consisting in operating several denoising parallel tracks.
- Can the authors comment on how credit assignment is learned/solved in their approach compared to DDPO? Since OGPO is only as farsighted as its critic, it seems crucial to discuss the the effective horizon length of the tasks tackled and how OGPO fairs against DDPO-like methods on those tasks.

**Limitations:**

yes

**Strengths And Weaknesses:**

Strengths:
- Every claim is supported by either 1+ reference(s) to prior art or to an appendix in the paper: very good.
- The descriptions of past works etc. are highly appreciated; those have been of great value throughout the review process.
- The LLM analogy with context and response: I think the analogy is overly stretched in this context and not helpful.
- I find the 2-subcolumn table valuable.
- “We do not compare to ReinFlow (Zhang et al., 2025b) due to reported reduced sample efficiency compared to DPPO, making the latter a more compelling baseline.” Since OGPO is based on DDPO with flow matching, i.e. ReinFlow, the argument above is not a convincing dismissal.


Weaknesses:
- Title: “Full-Finetuning” is a strange phrasing, or at least a phrasing I am unfamiliar with. To be honest I have no idea what “full” would refer to (before reading, a priori).
- Abstract: too many details about the method, and overall too long; out of place. Also: remove emphasis/italics.
- Highlighting every occurrence of OGPO is of poor taste. I suggest the authors refrain from doing that.
- Fig 1: it is unclear what the big red arrow represents. It seems easier, since Fig is a conputational graph, to tie Q in this graph, despite 0-th order.
- The story would benefit from a simplification; it takes a lot of effort to grasp what the authors intentions are with OGPO.


Minor:
- L19-20, 2nd col: “ask is: can” use a column.
- L27-28, 2nd col: emdash in LaTeX not type-setting well; correct emdash type-setting there.
- Same as above @ L50-51, @ L97-98
- L111, 1st col: clumsy notation.
- L158-159: “OGPO: On-Policy PPO for Off-Policy Policy Extraction.” make it into a sentence. It could act as a paragraph title if it were the acryonym.
- L191-192, 1st col: “lexicographically” the adverb can probably be dropped, unless there is a real ambiguity that I am missing.
- L187-190, 2nd col: no need here to type-set baseline and advantage with emphasis.
- Fig 4: should appear after the text where it is mentioned.

---

> ### Author Rebuttal · Authors · 2026-03-31
>
> We thank Reviewer T34B for the thorough, insightful review. We address each concern below.
>
> **Q1: Why emphasize zero-th order optimization over backpropagation through denoising steps?**
>
> There are two advantages to using zero-th order optimization: Zero order optimization through the *denoising process* avoids backpropagation (first-order optimization) through *denoising steps* as the ‘time’ axis (denoted as BPTT). BPTT exhibits vanishing/exploding gradients, and is ablated against in the Appendix Sec. J (and also attested as the DQL baseline in [2]). Zero-th order optimization does not need to backprop through the denoising steps.
>
> Our zero order optimization performs a policy update with respect to the critic Q(s,a) using importance sampling, whereas first order updates backpropagate over the entire GCP via Q(s,a). In many contact-rich settings, Q(s,a) can be non-smooth with respect to the “a” input, so BPTT can lead to unreliable gradients. [3] shows that zero-order optimization is preferable in these settings.
>
> To clarify these two benefits of zero-order optimization, we are adding an additional ablation which avoids BPTT but uses first-order gradients to optimize Q(s,a) as described in Flow Q-Learning (FQL). Together, the BPTT variant of OGPO and FQL will isolate the benefits from avoiding BPTT versus avoiding Q gradients.
>
> **Q2: Brittle wrt several denoising parallel tracks?**
> PPO-style methods require an estimate of the advantage A(s,a) = Q(s,a) - V(s). Our parallel tracks are a natural means of obtaining an estimate of $V(s) = \mathop{\mathbb{E}}_{a \sim \pi(s)} Q(s,a)$, without requiring a separate function approximator V(s), while allowing us to *reuse* those parallel trajectories for gradient computation. Using fewer tracks either (a) requires a separate “V” network (which may be uncalibrated for the Q networks in the ensemble) or (b) reduces accuracy in our V(s) estimate. At the same time, parallel tracks also provide more gradient signal across denoising paths.
>
> In the revision, we will ablate to measure the necessity of parallel track gradients: we will use 32 parallel tracks (default setting) to estimate $V(s)$, but compute gradients with 4, 8, 16 parallel tracks (and suitably normalize). We will ablate on # of parallel tracks v.s. UTD ratio as well. We agree that understanding the need for parallel tracks is an interesting question and worthy of future study.
>
> **Q3: Can the authors comment on credit assignment compared to DDPO?**
> We assume the reviewer is referring to DPPO [2] . There, the effective horizon is the product of the number of denoising steps and task horizon. The on-policy formalism precludes sample reuse from past suboptimal trajectories, and credit assignment is only given based on full-trajectory return. In contrast, OGPO learns a critic function, which enables efficient credit assignment to each (s,a) pair. Thus, in DPPO, a given (s,a) pair will be “good” only if it is part of a successful trajectory, whereas in OGPO, a (s,a) pair may be scored “good” by the critic even if the policy makes mistakes at later time-steps that harm full-trajectory return. We will include these clarifications in the revision. In DDPO [1], there is no environmental MDP, so credit assignment is purely computational.
>
> [1] Black, Kevin, et al. "Training diffusion models with reinforcement learning." (2023).
>
> [2] Ren, Allen Z., et al. "Diffusion policy policy optimization." (2024).
>
> [3] Suh, Hyung Ju, et al. "Do differentiable simulators give better policy gradients?." (2022)
>
> **W1: “Full-Finetuning” phrasing**
>
> “Full fine-tuning” refers to fine-tuning the entirety of the GCP process learning residual corrections [EXPO] or initial noise steering vectors [DSRL]. We will change it to “Full-Policy Finetuning” to clarify that “Full” refers to the entire policy, rather than residual or initial noise steering.
>
> **W2: Abstract too long**
>
> We will remove the italics and shorten the abstract in the revised manuscript and welcome further suggestions to improve its succinctness.
>
> **W3: Highlighting every occurrence of OGPO is of poor taste**
>
> We will remove method coloring (including for baselines) from the text until the experiments section. We will also clarify that method colors correspond to the plot colors, explaining our rationale for method coloring in the experiments.
>
> **W4: Fig 1: it is unclear what the big red arrow represents**
>
> Please see MM7P’s response Q1.
>
> **W5: The story would benefit from a simplification.**
>
> We will restructure the opening of Section 3 to lead with the core insight—that denoising trajectories are computationally cheap and can be sampled in parallel, enabling on-policy PPO updates over the denoising MDP while maintaining off-policy data reuse on the expensive environment MDP. We welcome specific suggestions for further simplification.
>
> **Minor comments**
>
> We thank the reviewer for all their careful observations. All minor comments will be fixed in the revised manuscript.

---

> > ### Author Rebuttal · Reviewer_T34B · 2026-04-03
> >
> > I thank the authors for the clarifications.
> >
> > I still believe that the computational burden remains a significant limitation, and in my view it continues to justify the score I initially assigned.

---

> > > ### Author Response · Authors · 2026-04-07
> > >
> > > Thank you for the prompt response and additional feedback! We address the computation burden below:
> > >
> > >
> > > **On computational burden as a limitation**
> > >
> > > We agree that OGPO’s per-step cost is higher than the baselines we compare against. However, in real-world robot learning settings, the core domain this paper aims to target, **sample efficiency** is the most important metric due to the wall-clock time needed for real-world environment steps and resets.
> > >
> > > On that axis, to achieve $\geq 95$\% success rate on a task like Robomimic Square using the MH dataset, OGPO takes only $\sim 0.75$ million steps, whereas DPPO takes $\sim8.5$ million steps, and DSRL takes $\sim 4$ million steps. OGPO is **11x** more sample-efficient than DPPO and **5.3x** more sample-efficient than DSRL. QC and EXPO cannot even reach a success rate of $\geq 95$\%.
> > >
> > > Moreover, most baselines fail to learn on more challenging (precise/bimanual and long-horizon) tasks such as Robomimic Transport and ToolHang, suggesting that increased per-step complexity may be *essential* for hard tasks. If OGPO requires 2–8x more compute per step but can fundamentally solve tasks that few baselines solve, we deem the trade-off worthwhile.
> > >
> > > **Scientific Merit**
> > >
> > > In addition to the method we propose, this paper has scientific merit by showing that full policy finetuning, when done correctly, is fundamentally more capable than other methods, not just in terms of sample efficiency but also in final task performance. Much of our contributions focus on extensive ablations that reveal the mechanisms by which full policy finetuning outperforms the other baselines at convergence.
> > >
> > > For the camera-ready version, we will also include additional experiments in which we learn a value function ($V(s)$); preliminary experiments show that this enables fewer additional rollouts for the advantage computation, thereby further reducing computational complexity.
> > >
> > > **Concrete Computational Statistics**
> > >
> > > However, even if computation is still a concern, we provide the following demonstration for each algorithm’s actor and critic update and find that:
> > >
> > > On an NVIDIA 5090 GPU, an online RL update for RM-Square with a batch size of 256 takes the following times:
> > >
> > > - OGPO: 40ms
> > > - EXPO: 15ms
> > > - QC: 16ms
> > > - DSRL: 8ms
> > > - DPPO: 6ms
> > >
> > > Most real-world robot learning pipelines run low-level controllers at $\leq 20$Hz, i.e., each action takes at least 50ms. With an action chunk of 4, the rollout will take 200ms. Our code uses Jax to facilitate multiprocessing, running gradient updates in parallel with the rollouts. Sans the use of extremely high update-to-data (UTD) ratios, smaller gradient update steps will result in plentiful GPU idle time across all algorithms supported by our codebase. Hence, sample efficiency is the key bottleneck for performance.
> > >
> > > **Image-Based Runs**
> > >
> > > Finally, we’re including a new range of VLA experiments from pixels where we use the VLA’s frozen VLM backbone as a visual representation. In these experiments, we pre-cache the VLM encoder, which is a substantial driver for the cost at parallel inference. In this setting, policy and critic updates remain the same complexity despite using more powerful vision encoders. This is a proof-of-concept for the method's scalability:
> > >
> > > Wall-clock time for convergence ($\geq 90\%$ success rate) with OGPO-VLM:
> > >
> > > - Square (1 camera): 8 hrs (650k steps)
> > > - Toolhang (1 camera): 19 hrs (1.23 million steps)
> > > - Transport (2 cameras): 36 hrs (1.5 million steps)

---

### Official Review · Reviewer_MM7P · 2026-03-16

**Soundness:** 3
**Presentation:** 2
**Significance:** 3
**Originality:** 3
**Overall Recommendation:** 3
**Confidence:** 3

**Summary:**

The authors introduce a method to improve the sample-efficiency of Generative Control Policies (GCPs), with a focus on flow matching GCPs (although they explain in the appendix how it can be extended to diffusion-based GCPs.).
They propose to do full finetuning of off-policy GCPs. Their training process alternates between off-policy Q learning using a TD-learning objective, and then policy optimization. For policy optimization, they have a bilevel optimization problem between denoising steps and environment steps. They use a modified PPO objective, which they finetune on denoising trajectories only, and this can be done by sampling actions from the replay buffer, and it can be done in parallel, which is very sample-efficient compared to previous approaches that do BPTT. They introduce two variants of their method, OGPO and OGPO+.
They conduct extensive experiments and ablations, and they show that their method outperforms various baselines using various RL paradigms, with major gains in sample-efficiency.

**Compliance With Llm Reviewing Policy:**

Affirmed.

**Key Questions For Authors:**

1. Do the arrows in Fig. 1 have a meaning?
2. In Section 4. Experiments 4, Table 1, in the Dense/dexterous setting: why are task-specific hyper parameters worse than fixed hyperparameters across tasks?
3. In Fig. 3: why do EXPO and EXPO+ perform so differently on some tasks (e.g., one reaches the highest SR among baselines while the other reaches the lowest.)? Why does performance drop for EXPO with more steps? In Fig. 4: what happens if OGPO+ is kept running?

**Limitations:**

yes

**Strengths And Weaknesses:**

**Strengths**
- Novel algorithm which effectively combines off-policy critic learning (sample-efficient) with on-policy PPO updates on the denoising MDP (expressive policy updates).
- Clear preliminaries to give the necessary background on GCPs, flow-based policies, and the bilevel MDP formulation they employ.
- The authors conduct extensive experiments across diverse manipulation tasks. The choice of baselines is comprehensive and covers all major paradigms related to their method (on-policy, partial fine-tuning, behavior cloning). Their method achieves 7x sample efficiency improvements over the DPPO baseline and reaches success rates near 100%.
- Precise implementation details to facilitate reproducibility.


**Weaknesses**
- The method description is very hard to follow. It is hard to understand it by reading only the main paper. Many important method details are deferred to the appendix.
- The distinction between training the base policy and the finetuning stage is not obvious in the method section, it requires further explanation.
- The distinction between OGPO and OGPO+ is confusing. Both seem to be essentially the same method, with one including some additional regularization. Further differences between the two methods only appear in Algorithm 2 in the appendix.
- Some experimental details should appear in the main section, at least the most important ones, such as the types of tasks that are considered, the state spaces, the reward structure and the distinction between base policy training and finetuning.
- Images of the environments are missing, and qualitative evaluations are not provided.


**Minor comments and typos**
- “action gradients with respect to Qtarg” in Section 3., paragraph OGPO: shouldn’t it be the other way around?
- In Section 3., OGPO paragraph: an bi-level MDP → a bi-level MDP.
- “The term V chosen in a manner…” “is” missing.
- In 4. “We describe limitations of our approach in?”: missing section.
- Fig. 11 is too small.

---

> ### Author Rebuttal · Authors · 2026-03-31
>
> We thank Reviewer MM7P for the careful and constructive review. We appreciate the reviewers favorable assessment of the novelty of our algorithm, the clarity of our preliminaries and bi-level MDP formulation, and our comprehensive experimental comparisons. Here, we will clarify the most salient questions.
>
>
> **Q1: Do the arrows in Fig. 1 have a meaning?**
>
> The filled-tip arrows demonstrate the computational graph of the denoising MDP rollout, while the hollow-tip arrows demonstrate transitions in the environmental MDP rollout. Finally, the red arrow for the Q function indicates OGPO’s ability to use Q-functions at each environment step to perform on-policy updates on the GCP. In contrast, pure on-policy methods require multiple environment rollouts before updating policies, making them sample inefficient. We will add these clarifications, along with a legend for the arrows, to the final manuscript.
>
> **Q2: In Table 1, Dense/Dexterous setting: why are task-specific hyperparameters worse than fixed hyperparameters?**
>
> Good catch! Previously, we chose 1 set of fixed hyperparameters representing the best across all benchmarks, per method; we then chose task-specific parameters that were mutually exclusive to the best fixed hyperparameters. We now recognize this is unintuitive, and set task-specific to denote maximum performance of “best-fixed” and other task-specific settings.
>
> **Q3.1 Why do EXPO and EXPO+ perform so differently on some tasks?**
>
> This is a great question. We hypothesize that the reason for this degradation is that changing the base policy, as occurs in SFT, creates non-stationarity for the inputs to the EXPO residuals, degrading performance. We will describe this hypothesis in the main text. Furthermore, we will include “+” versions of UMAP plots in the Appendix Fig. 12 and show how adding the success buffer training increases non-stationarity.
>
> **Q3.2 Why does performance drop for EXPO with more steps?**
>
> During RL training, EXPO's residual policy, trained with SAC, aggressively over-optimizes the Q functions due to high update-to-data (UTD) ratios, leading the policy into a local maxima from which it becomes hard to escape despite further training. We verified this via a grid sweep on EXPO, spanning UTD 1-20, and reported the experiment that yielded the highest performance. We will describe these ablations in the revision Sec 4.1.
>
> **Q3.3 In Fig. 4: what happens if OGPO+ is kept running?**
>
> We have trained OGPO+ on over a million steps (~6000 episode rollouts) post-success without collapse. The policy simply asymptotically oscillates between 95-100% success rate for square, and 85-95% success rate for transport. We will include a 2 million step run in Fig. 4 of the final manuscript.
>
> **W1: Method description hard to follow; details deferred to appendix**
>
> We appreciate this feedback. We plan to move the details about the policy gradient loss and the bi-level MDP to the body to improve clarity. Please let us know if there are any other details you would like us to move to the main text so that we can incorporate your feedback.
>
> **W2: The distinction between training the base policy and the finetuning stage is not obvious**
>
> We define a *base policy* as a pre-trained behavior cloning policy, which is trained via the same behavior cloning loss. We will clarify our terminology in the revision. We have added a new “Pre-training” paragraph under Preliminaries to the draft to clarify these details, and explain how the pre-training learns a base policy and initializes the Q-functions. We consider these details separate from the OGPO algorithm, and are held fixed across baselines.
>
> **W3: The distinction between OGPO and OGPO+ is confusing**
>
> OGPO+ consists of simple algorithmic modifications to OGPO to improve performance, incorporating best practices from prior work. Because it only subtly modifies OGPO, we call it OGPO+. Comparing both methods isolates what and how these additional modifications improve performance. We will clarify our choice of nomenclature at the beginning of Sec 3.1
>
> **W4: Some important experimental details should appear in the main section**
>
> Thank you for this suggestion – due to the additional permitted space, we are now able to fit additional experiment details from the Appendix to Sec. 4 “Experiments” paragraph.
>
> **W5: Images of the environments are missing, and qualitative evaluations are not provided.**
>
> We will add example environment images below Figure 3, a pointer to the existing qualitative evaluations of ToolHang in Appendix Figure 13, and additional qualitative evaluations in more environments.
>
> **Minor comments and typos**
>
> We thank the reviewer for all these careful observations. All minor comments and typos will be fixed in the revised manuscript

---

> > ### Author Rebuttal · Reviewer_MM7P · 2026-04-04
> >
> > Thank you for the rebuttal and for providing more details. I appreciate all the clarifications, but after re-reading the paper, I still believe that too many important details only appear in the appendix, making it really hard to understand the method from the main paper only, and I don't think everything can be fit in the main paper.

---

> > > ### Author Response · Authors · 2026-04-07
> > >
> > > Thank you for your concerns; we truly appreciate your attention to detail and the high standards you hold the paper to. Regardless of the acceptance decision, we would love to improve upon the manuscript and incorporate your feedback. Could you please let us know which appendix details are missing from the main paper that made the method difficult to understand? We are trying to understand how we can improve the quality of our exposition in the hopes of reaching the broadest possible audience.

---

### Decision · Program_Chairs · 2026-04-30

**Decision:**

Accept (regular)

**Comment:**

The paper proposes OGPO, a decoupled framework for sample-efficient fine-tuning of generative control policies by combining off-policy critic learning on the environment MDP with on-policy PPO updates on the denoising MDP.  The proposed sample efficient fine-tuning approach is novel and interesting. The conceptual separation of the physically expensive environmental MDP (solved via off-policy Critic) and the computationally cheap denoising MDP (solved via on-policy PPO) is an elegant and highly practical solution.  The idea has been tested across a lot of popular robotic benchmarks. Some limitations are noted around computational overhead due to extensive parallel denoising rollouts, which weakens the practical efficiency claims beyond environment interactions. Overall, a novel contribution.